# APE1 recruits ATRIP to ssDNA in an RPA-dependent and -independent manner to promote the ATR DNA damage response

Yunfeng Lin[1], Jia Li[1], Haichao Zhao[1], Anne McMahon[1], Kelly McGhee[1], Shan Yan[1,2,3]*

[1]Department of Biological Sciences, University of North Carolina at Charlotte, Charlotte, United States; [2]School of Data Science, University of North Carolina at Charlotte, Charlotte, United States; [3]Center for Biomedical Engineering and Science, University of North Carolina at Charlotte, Charlotte, United States

**Abstract** Cells have evolved the DNA damage response (DDR) pathways in response to DNA replication stress or DNA damage. In the ATR-Chk1 DDR pathway, it has been proposed that ATR is recruited to RPA-coated single-stranded DNA (ssDNA) by direct ATRIP-RPA interaction. However, it remains elusive how ATRIP is recruited to ssDNA in an RPA-independent manner. Here, we provide evidence that APE1 directly associates ssDNA to recruit ATRIP onto ssDNA in an RPA-independent fashion. The N-terminal motif within APE1 is required and sufficient for the APE1-ATRIP interaction in vitro and the distinct APE1-ATRIP interaction is required for ATRIP recruitment to ssDNA and the ATR-Chk1 DDR pathway activation in *Xenopus* egg extracts. In addition, APE1 directly associates with RPA70 and RPA32 via two distinct motifs. Taken together, our evidence suggests that APE1 recruits ATRIP onto ssDNA in an RPA-dependent and -independent manner in the ATR DDR pathway.

*For correspondence:
shan.yan@uncc.edu

Competing interest: The authors declare that no competing interests exist.

## Editor's evaluation

This important paper provides new insight into the mechanism of the activation of DNA damage checkpoint (DDR) in response to the single-stranded DNAs (ssDNAs). The authors used *Xenopus* egg extracts and a reconstitution reaction with purified proteins and presented convincing results to support the authors' claims on a non-catalytic role of APE1 endonuclease to recruit DDR activator, ATRIP, to the ssDNA for DDR activation. The work would be of interest to researchers who work on the cell cycle and DNA damage responses as well as DNA repair.

## Introduction

The DNA damage response (DDR) signaling pathways such as ATR-Chk1 and ATM-Chk2 are activated by DNA replication stress or different DNA damage to coordinate DNA repair with cell cycle as well as apoptosis and senescence (*Bartek et al., 2004*; *Branzei and Foiani, 2010*; *Ciccia and Elledge, 2010*; *Cimprich and Cortez, 2008*; *Harper and Elledge, 2007*; *Harrison and Haber, 2006*; *Su, 2006*). In response to stalled DNA replication forks and different DNA lesions including DNA double-strand breaks (DSBs) and single-strand breaks (SSBs), ATR DDR can be recruited to and activated by RPA-coated single-stranded DNA (ssDNA) derived from functional uncoupling of MCM helicase and DNA polymerase activities, DSB end resection in the 5'–3' direction, or SSB end resection in the 3'–5' direction (*Ciccia and Elledge, 2010*; *Cimprich and Cortez, 2008*; *Lin et al., 2018*; *Maréchal and Zou, 2015*; *Shiotani and Zou, 2009*). ATR activation also requires several mediator proteins such as ATRIP,

TopBP1, and the Rad9-Rad1-Hus1 (9-1-1) complex (*Delacroix et al., 2007*; *Kumagai et al., 2006*; *Yan and Michael, 2009*; *Zou and Elledge, 2003*). Activated ATR kinase then phosphorylates a variety of substrates such as Chk1, among others, and phosphorylated Chk1 is the activated version of Chk1 kinase to regulate cell cycle progression and often serves as an indicator of ATR DDR activation (*Chen and Sanchez, 2004*; *Guo et al., 2000*; *Matsuoka et al., 2007*).

Since the discovery of ATRIP in ATR DDR pathway about 20 years ago, it has been an active subject of studies regarding how the ATR-ATRIP complex is recruited to ssDNA and activated by ATR activator/mediator proteins to maintain genome integrity (*Cortez et al., 2001*; *Zou and Elledge, 2003*). Earlier studies from several groups using human and yeast cells as well as *Xenopus* egg extracts have revealed independently that ATR and ATRIP associate with each other into a tight complex and that the direct ATRIP recognition and interaction with RPA-ssDNA is essential for the recruitment of ATR to ssDNA regions at sites of DNA damage for ATR activation (*Lee et al., 2003*; *Unsal-Kaçmaz and Sancar, 2004*; *Zou and Elledge, 2003*). However, the RPA requirement of ATRIP recruitment to ssDNA for ATR DDR activation is sort of questioned by several follow-up reports demonstrating that ATR-ATRIP complexes can bind to ssDNA in an RPA-independent manner in vitro, and that the low affinity RPA-independent recruitment of ATRIP to ssDNA is mediated by an unknown protein in mammalian cell nuclear extracts (*Bomgarden et al., 2004*; *Kim et al., 2005*). Although the exact molecular determinants of ATRIP (such as the N-terminal 1–108 amino acid fragment of human ATRIP) interaction with RPA-ssDNA remain to be determined and reconciled (*Ball et al., 2005*; *Namiki and Zou, 2006*), additional lines of investigations have demonstrated that TopBP1 can directly activate the ATR-ATRIP complex in *Xenopus* egg extracts and reconstituted human proteins in an RPA-dependent and RPA-independent manner (*Choi et al., 2010*; *Choi et al., 2007*; *Kumagai et al., 2006*). Whereas a good progress has been made regarding the implication of post-translational modifications such as sumoylation and phosphorylation in ATRIP regulation in ATR DDR (*Memisoglu et al., 2019*; *Wagner et al., 2019*; *Wu et al., 2014*), it remains an outstanding question in the field of genome integrity of how exactly ATRIP is recruited to ssDNA in an RPA-dependent and/or -independent fashion for ATR DDR activation.

As the major AP endonuclease, AP endonuclease 1 (APE1) has fast AP endonuclease activity but slow 3'–5' exonuclease and 3'-phosphodiesterase activities as well as 3'–5' RNA phosphatase and exoribonuclease activities (*Boiteux and Guillet, 2004*; *Burkovics et al., 2006*; *Chohan et al., 2015*; *Hadi et al., 2002*; *Tell et al., 2009*; *Wilson and Barsky, 2001*). In addition to its role in redox regulation for transcription, APE1 plays essential roles in various DNA repair pathways (*Li and Wilson, 2014*; *Tell et al., 2009*). Whereas APE1-knockout mice are embryonic lethal, the underlying mechanism of APE1 in cell viability remains unclear (*Fung and Demple, 2005*; *Masani et al., 2013*; *Xanthoudakis et al., 1996*). Human APE1 is genetically altered and aberrantly expressed and localized in cancer patients and has become an emerging therapeutic target for various cancer therapy (*Abbotts and Madhusudan, 2010*; *Al-Safi et al., 2012*; *Fishel and Kelley, 2007*; *Koukourakis et al., 2001*; *Sengupta et al., 2016*; *Thakur et al., 2014*; *Yoo et al., 2008*). Recent pre-clinical and clinical studies have shown encouraging finding of APE1 inhibitor APX3330 in anti-cancer therapy in solid tumors (*Caston et al., 2021*; *Shahda et al., 2019*). Our recent studies have demonstrated that the ATR-Chk1 DDR pathway is activated by hydrogen peroxide-induced oxidative DNA damage and defined plasmid-based SSB structures in *Xenopus* high-speed supernatant (HSS) system (*Lin et al., 2018*; *Wallace et al., 2017*; *Willis et al., 2013*). To promote the ATR DDR activation, APE1 initiates the 3'–5' end resection at SSB sites to generate a short ~1–3 nt-ssDNA gap via its exonuclease activity, followed by PCNA-mediated APE2-dependent SSB end resection continuation (*Lin et al., 2018*; *Lin et al., 2020*). This APE1/2-mediated two-step processing of SSBs generates a longer stretch of ssDNA (~18–26 nt) coated by RPA, leading to the assembly of the ATR DDR complex (ATR-ATRIP, TopBP1, and 9-1-1 complex), subsequent ATR DDR activation, and eventual SSB repair (*Hossain et al., 2018*; *Lin et al., 2018*; *Lin et al., 2020*). However, it remains unknown whether APE1 plays a direct role in the ATRIP recruitment to ssDNA via a non-catalytic function in the presence and/or absence of RPA for the ATR DDR pathway.

Here, we provide direct evidence that in addition to its exonuclease-mediated function, APE1 plays a direct role in the recruitment of ATRIP to ssDNA in *Xenopus* egg extracts and in in vitro reconstitution system with purified proteins. The N-terminal motif of APE1 is required for its direct association with ssDNA in vitro, and such APE1-ssDNA interaction can be enhanced by RPA. Importantly, APE1

directly interacts with ATRIP and recruits ATRIP to ssDNA with the absence of RPA in vitro. A mutant APE1 deficient for ATRIP interaction but proficient for ssDNA association could not recruit ATRIP onto ssDNA in the presence of endogenous RPA in APE1-depleted *Xenopus* egg extracts. Similar to wild type APE1, a nuclease mutant APE1 still recruits ATRIP onto ssDNA in APE1-depleted HSS, suggesting that APE1's role in ATRIP ssDNA recruitment is not dependent on its nuclease activity. APE1 directly associates with RPA in vitro via two distinct motifs within APE1. Notably, the RPA-interaction-deficient APE1 had no effect on the ATRIP recruitment onto ssDNA in *Xenopus* egg extracts. Overall, the data in this study demonstrating that APE1 is required for ATRIP recruitment to RPA-coated ssDNA for ATR DDR activation in *Xenopus* egg extracts, and that APE1 directly associates with and recruits ATRIP to ssDNA in the absence of RPA in vitro. These findings thus support a previously uncharacterized critical non-catalytic function of APE1 for direct ATRIP recruitment to ssDNA independent of RPA for the ATR DDR pathway.

## Results

### APE1 is required for the recruitment of ATRIP onto ssDNA in the ATR-Chk1 DDR pathway activation in *Xenopus* egg extracts

Our recent studies have revealed that APE1 plays an important role in the defined SSB structure-induced ATR-Chk1 DDR pathway via its 3'–5' exonuclease activity in *Xenopus* HSS system (*Lin et al., 2018*; *Lin et al., 2020*). This APE1-mediated initiation of 3'–5' SSB end resection (~1–3 nt-ssDNA gap) will be followed by APE2 recruitment and activation to continue SSB end resection, generating a longer stretch of ssDNA (~18–26 nt-ssDNA) coated by RPA for subsequent assembly of an ATR check-point protein complex including ATR-ATRIP, TopBP1, and 9-1-1 complex to activate ATR DDR (*Lin et al., 2018*; *Wallace et al., 2017*; *Willis et al., 2013*). In addition to the 3'–5' SSB end resection initiation, we are interested in whether APE1 plays other roles such as non-catalytic function in the ATR DDR pathway. To test this question directly, we chose to utilize a defined plasmid DNA with a 1–3 nt small ssDNA gap structure (designated as Gap plasmid) in vitro as previously described (*Lin et al., 2018*), and tested whether APE1 is still required for the ATR DDR in response to this Gap plasmid in HSS (top of the panel, *Figure 1A*). As expected, the defined Gap plasmid ('Gap'), but not control plasmid ('CTL') triggered Chk1 phosphorylation, suggesting the activation of the ATR-Chk1 DDR pathway by the defined Gap plasmid in the *Xenopus* HSS (*Figure 1A*). Notably, the Gap plasmid-induced Chk1 phosphorylation was still impaired in APE1-depleted HSS system ('extract', *Figure 1A*), suggesting that APE1 may play an additional non-catalytic function in the ATR DDR pathway. To further define the additional role of APE1 in ATR DDR activation, we isolated the DNA-bound fractions from the HSS and examined the abundance of checkpoint proteins via immunoblotting analysis. Although the recruitment of RPA70 and RPA32 to the Gap plasmid was slightly reduced when APE1 was depleted from the HSS ('DNA-bound', *Figure 1A*), the presence of RPA70 and RPA32 on Gap-plasmid DNA suggests that the gap structure can be further processed by APE2 in the APE1-depleted HSS, consistent with our previous studies (*Lin et al., 2018*; *Lin et al., 2020*). Notably, the recruitment of ATRIP onto Gap plasmid was significantly compromised in APE1-depleted HSS, suggesting that APE1 plays a more direct role in ATRIP recruitment to ssDNA regions in the defined Gap plasmid (*Figure 1A*).

In light of different lengths of ssDNA in vitro reconstitution systems by previous studies (such as 75 nt or 80 nt) (*Bomgarden et al., 2004*; *Choi et al., 2010*; *Zou and Elledge, 2003*), we intended to study the recruitment of ATRIP onto defined double-stranded DNA (dsDNA) structures with different length of ssDNA gaps. We chose to test two 100 bp-dsDNA structures with either 30 nt- or 80 nt-ssDNA gap covalently linked with 5'-biotin on top strand (designated as '30 nt gap' and '80 nt gap') for subsequent streptavidin magnetic bead-bound isolation and analysis from *Xenopus* egg extracts (top of the panel of *Figure 1B*). When equal moles of dsDNA with 30 nt or 80 nt-ssDNA gap were added to HSS, more RPA70 and RPA32 as well as APE1 and ATRIP are recruited to the 80 nt-ssDNA gap and Chk1 phosphorylation was also enhanced (*Figure 1B*). This enhanced Chk1 phosphorylation is likely due to increased RPA complex recruitment onto the 80 nt-ssDNA gap (*Figure 1B*). Whereas APE1 depletion led to compromised Chk1 phosphorylation, the recruitment of ATRIP but not RPA70 nor RPA32 was compromised in APE1-depleted HSS (*Figure 1B*). Our observations so far suggest that APE1 is important for the recruitment of ATRIP onto RPA-coated ssDNA and Chk1 phosphorylation in *Xenopus* HSS.

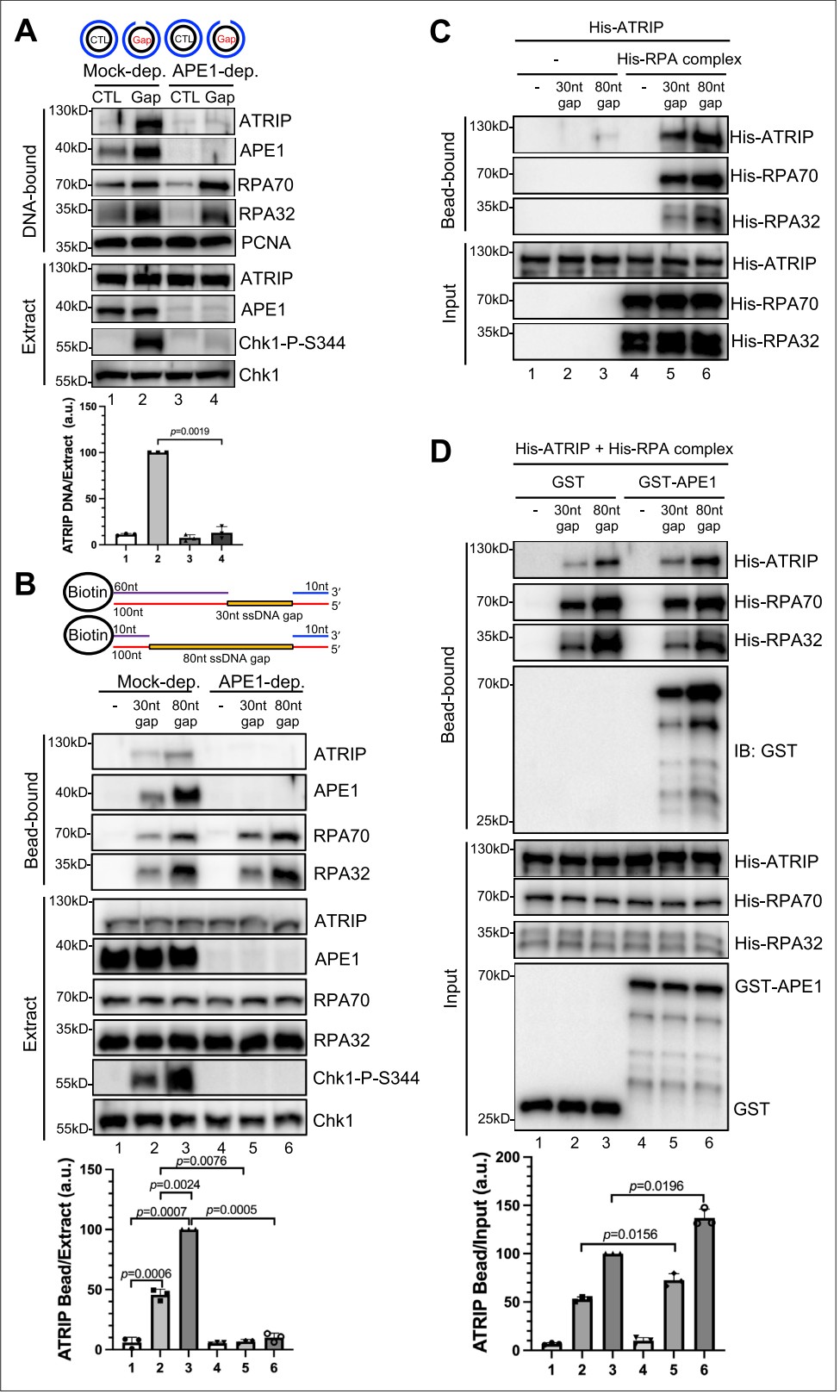

**Figure 1.** AP endonuclease 1 (APE1) is required for the recruitment of ATRIP to RPA-coated single-stranded DNA (ssDNA) in *Xenopus* egg extracts. (**A**) CTL (control) or Gap plasmid was added to Mock- or APE1-depleted high-speed supernatant (HSS) and incubated for 30 min. The DNA-bound fractions and total egg extract were examined via immunoblotting analysis as indicated. (**B**) Streptavidin beads coupled with equal moles of biotin-

*Figure 1 continued on next page*

*Figure 1 continued*

labeled double-stranded DNA (dsDNA) with ssDNA gap structures (30 nt or 80 nt) were added to Mock- or APE1-depleted HSS. After incubation for 30 min at room temperature, the DNA-bound fractions and total egg extract were examined via immunoblotting analysis as indicated. (**C**) Streptavidin beads coupled with equal moles of biotin-labeled dsDNA with ssDNA gap structures (30 nt or 80 nt) were added to an interaction buffer containing purified His-ATRIP protein with/without His-RPA protein. After incubation for 30 min at room temperature, the DNA-bound fractions and the input were examined via immunoblotting analysis. (**D**) Streptavidin beads coupled with equal moles of biotin-labeled dsDNA with ssDNA gap structures (30 nt or 80 nt) were added to an interaction buffer containing His-ATRIP and His-RPA, which was supplemented with GST or GST-APE1. After incubation for 30 min at room temperature, the DNA-bound fractions and the input were examined via immunoblotting analysis. (**A, B, D**) ATRIP intensity was quantified, and the ratio of ATRIP from DNA-bound vs extract/input was examined. a.u., arbitrary unit. Mean ± SD, n=3.

The online version of this article includes the following source data and figure supplement(s) for figure 1:

**Source data 1.** Raw images of immunoblotting analysis referenced in *Figure 1A*.

**Source data 2.** Raw images of immunoblotting analysis referenced in *Figure 1B*.

**Source data 3.** Raw images of immunoblotting analysis referenced in *Figure 1C*.

**Source data 4.** Raw images of immunoblotting analysis referenced in *Figure 1D*.

**Figure supplement 1.** AP endonuclease 1 (APE1) further promote ATRIP's recruitment to RPA-coated DNA.

**Figure supplement 1—source data 1.** Raw images of immunoblotting analysis referenced in *Figure 1—figure supplement 1A*.

**Figure supplement 1—source data 2.** Raw images of immunoblotting analysis referenced in *Figure 1—figure supplement 1B*.

**Figure supplement 1—source data 3.** Raw images of immunoblotting analysis referenced in *Figure 1—figure supplement 1C*.

**Figure supplement 1—source data 4.** Raw images of immunoblotting analysis referenced in *Figure 1—figure supplement 1D*.

To re-evaluate the role of RPA in the recruitment of ATRIP onto ssDNA gaps, we performed RPA depletion experiment in HSS and characterized the phenotype of RPA depletion in ATRIP recruitment and ATR-Chk1 DDR pathway. With anti-RPA antibodies, majority of the endogenous RPA protein complex, if not all, was immunodepleted from HSS (extract panel, *Figure 1—figure supplement 1A*), and such RPA depletion led to almost no binding of RPA protein complex (RPA70 and RPA32) onto the defined ssDNA gaps and impairment of the ssDNA-induced Chk1 phosphorylation (*Figure 1— figure supplement 1A*). Whereas RPA deletion did not decrease the recruitment of APE1 onto ssDNA gaps, the recruitment of endogenous ATRIP protein onto ssDNA gaps (30 nt and 80 nt) was impaired in RPA-depleted HSS (bead-bound panel and quantification panel, *Figure 1—figure supplement 1A*). We noted that anti-RPA antibodies co-depleted some endogenous ATRIP protein from HSS, but had almost no co-depletion of endogenous APE1 protein in HSS (Lanes 1–3 vs Lanes 4–6 in extract panel, *Figure 1—figure supplement 1A*). This co-depletion of endogenous ATRIP protein is similar to a previous observation by the Dunphy group (*Kim et al., 2005*), and may suggest a tight complex formation between endogenous ATRIP and RPA protein complex in *Xenopus* egg extracts. Our quantifications of ssDNA-bound ATRIP normalized to available ATRIP protein in HSS showed that RPA depletion almost had no effect on ATRIP's binding to the 30 nt-ssDNA gap, and only mildly decreased the binding of ATRIP (~20%) to 80 nt-ssDNA gap (quantification panel, *Figure 1—figure supplement 1A*). These observations suggest that the recruitment of ATRIP onto ssDNA gaps is RPA-dependent and RPA-independent in the HSS system.

Because ssDNA gaps have been widely accepted as a central platform for protein recruitment and activation of the ATR/Chk1 DDR pathway (*Cimprich and Cortez, 2008*; *Maréchal and Zou, 2015*), we are prompted to evaluate the role of APE1 in the ATR/Chk1 DDR pathway in cultured cells. Hydrogen peroxide-induced oxidative DNA damage triggered Chk1 phosphorylation at Ser345 and Ser317 in human osteosarcoma U2OS cells, suggesting the activation of ATR DDR (*Figure 1—figure supplement 1B*). Notably, siRNA-mediated APE1 knockdown of human APE1 in U2OS cells compromised the hydrogen peroxide-induced Chk1 phosphorylation, suggesting that APE1 is important for the ATR/Chk1 DDR in cultured human cells (*Figure 1—figure supplement 1B*). Our observation here is

consistent with two prior studies showing that APE1 is important for the ATR DDR pathway activation in response to oxidative DNA damage in human cancer cells MDA-MB-231 and PANC1 cells, and ultraviolet damage in non-dividing nucleotide excision repair-deficient (i.e. XPC$^{-/-}$) cells (*Li et al., 2022*; *Vrouwe et al., 2011*).

Next, to determine whether APE1 plays any direct role in the RPA-dependent ATRIP recruitment onto ssDNA gaps, we tested whether His-tagged ATRIP recombinant protein can be recruited to 30 nt- or 80 nt-ssDNA gap in vitro in the absence or presence of equal moles of recombinant RPA complex. Consistent with previously reported RPA-dependent ATRIP recruitment to ssDNA (*Zou and Elledge, 2003*), 30/80 nt-ssDNA coated with RPA70 and RPA32 significantly enhanced the recruitment of His-ATRIP in vitro, although almost no binding of ATRIP onto ssDNA (30 nt and 80 nt) was observed in the absence of recombinant RPA complex (*Figure 1C*). We noticed more binding of His-ATRIP onto 80 nt-ssDNA gap compared with 30 nt-ssDNA gap (*Figure 1C*). As expected, the recruitment of His-ATRIP onto 30 nt- and 80 nt-ssDNA was similar to each when same amount of ssDNA gap structures was coupled to beads (*Figure 1—figure supplement 1C*). Furthermore, the addition of GST-APE1 but not GST protein increased the recruitment of His-ATRIP onto ssDNA with the presence of His-RPA complex in vitro (*Figure 1D*). It is worth noting that the presence of GST-APE1 had almost no noticeable effect on the recruitment of His-RPA70 and His-RPA32 to ssDNA gap structures (*Figure 1D*). Similarly, the presence of GST-APE1 but not GST increased the recruitment of endogenous ATRIP but not endogenous RPA70/RPA32 to ssDNA gap structures in the *Xenopus* HSS (*Figure 1—figure supplement 1D*). Whereas RPA itself is sufficient for ATRIP recruitment onto ssDNA in vitro, our observations here suggest that APE1 may stimulate the RPA-dependent ATRIP recruitment onto ssDNA in vitro. Alternatively, it is possible that APE1 may play an additional but direct role in the recruitment of ATRIP onto ssDNA in vitro that is independent of RPA.

## APE1 recognizes and binds with ssDNA directly in a length-dependent manner in vitro

Although APE1 is known as a DNA repair protein to specifically recognize and process AP site, it remains unclear whether and how APE1 interacts with ssDNA. To identify the possible direct role of APE1 in ATRIP recruitment onto ssDNA, we first performed systematic analysis of APE1 association with ssDNA. Our bead-bound experiments showed that GST-APE1 but not GST was recruited onto 30 nt- and 80 nt-ssDNA gap structures in vitro (*Figure 2A–B*). We also determined that GST-APE1 but not GST was recruited onto beads coupled with 70 nt-ssDNA in vitro (*Figure 2—figure supplement 1*). Furthermore, we demonstrated that GST-APE1 but not GST was recruited to beads coupled with 40 nt-, 60 nt-, and 80 nt-ssDNA, but not 10 nt- nor 20 nt-ssDNA (*Figure 2C*). Furthermore, the longer ssDNA is, the more GST-APE1 is recruited (*Figure 2C*). Collectively, these observations suggest an APE1-ssDNA interaction in a length-dependent manner in vitro (30–80 nt) regardless the ssDNA is alone or in gapped structures.

To further dissect domain requirements within APE1 for ssDNA association, we generated a series of deletion GST-tagged APE1 and found that WT GST-APE1 and AA101-316 GST-APE1 but not any other deletion GST-APE1 tested (i.e. AA35-316, AA1-100, AA1-34, AA35-100, AA101-200) associated with beads coupled with 70 nt-ssDNA in vitro (*Figure 2A and D*). Intriguingly, AA101-316 but not AA35-316 GST-APE1 associated with ssDNA (*Figure 2D*). We speculate that the fragment of AA35-100 within APE1 may somehow inhibit the APE1-ssDNA association due to a currently unknown mechanism. In addition, our electrophoretic mobility shift assays (EMSA) revealed that WT GST-APE1 but not GST formed protein-ssDNA complex in vitro (*Figure 2E*). Notably, neither AA35-316 nor AA1-34 GST-APE1 formed protein-ssDNA complex in EMSA (*Figure 2E*). These observations suggest that AA1-34 within APE1 is required but seems not sufficient for ssDNA association at least under our tested conditions, and that APE1 AA35-316 is deficient for ssDNA association while APE1 AA101-316 is proficient in ssDNA interaction (*Figure 2*).

What are the effects of N-terminal motif of APE1 for its 3'–5' exonuclease and AP endonuclease activities? Similar to our previous report (*Lin et al., 2020*), WT GST-APE1 but neither ED (E95Q-D306A) GST-APE1 nor GST displayed 3'–5' exonuclease and AP endonuclease activities (*Figure 2—figure supplement 2A–B*). Notably, AA101-316 GST-APE1 is defective for 3'–5' exonuclease and AP endonuclease activities (*Figure 2—figure supplement 2A–B*); however, AA35-316 GST-APE1 is proficient in AP endonuclease activity but deficient for 3'–5' exonuclease activity (*Figure 2—figure*

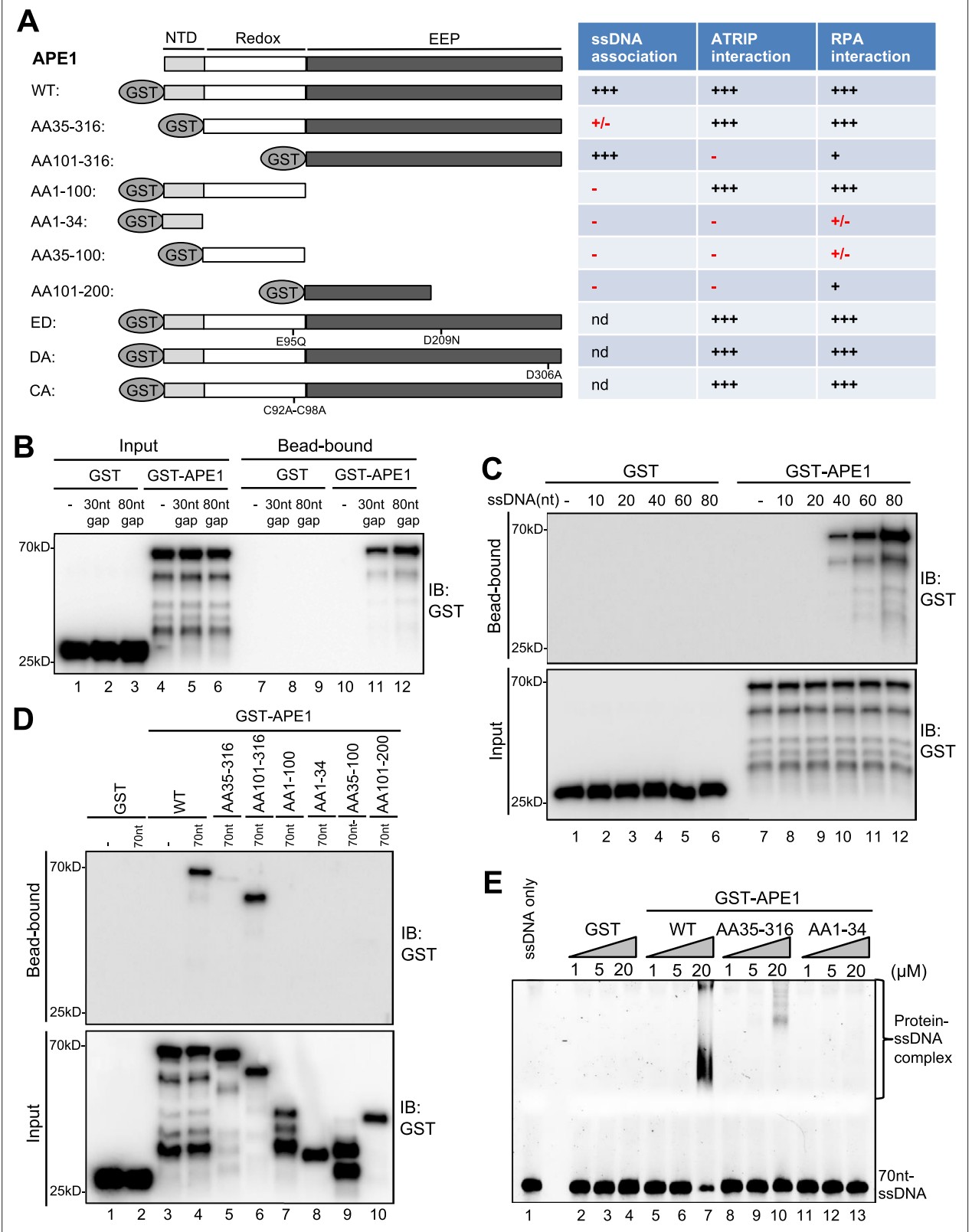

**Figure 2.** AP endonuclease 1 (APE1) recognizes and binds with single-stranded DNA (ssDNA) in a length-dependent fashion in vitro. (**A**) Schematic diagram of APE1 functional domains and a summary of its interactions with ssDNA, ATRIP, and RPA from this study. Various symbols indicate estimates of APE1 interactions: '+++', indicates the strongest interaction; '+' indicates moderate interaction; '+/-' indicates minimal to no interaction; '-' indicates almost no interaction; 'nd', not determined. (**B**) Streptavidin beads coupled with biotin-labeled double-stranded DNA (dsDNA) with ssDNA gap

*Figure 2 continued on next page*

*Figure 2 continued*

structures (30 nt or 80 nt) were added to an interaction buffer containing GST or GST-APE1. After incubation for 30 min at room temperature, the DNA-bound fractions and the input were examined via immunoblotting analysis as indicated. (**C**) Streptavidin beads coupled with biotin-labeled ssDNA with different lengths (10 nt, 20 nt, 40 nt, 60 nt, or 80 nt) were added to an interaction buffer containing GST or GST-APE1. After incubation for 30 min at room temperature, the DNA-bound fractions and the input were examined via immunoblotting analysis. (**D**) Streptavidin beads coupled with biotin-labeled ssDNA (70 nt) were added to an interaction buffer containing (70 nt) GST or WT or fragment of GST-APE1. After incubation for 30 min at room temperature, the DNA-bound fractions and the input were examined via immunoblotting analysis. (**E**) An electrophoretic mobility shift assay (EMSA) shows the interaction between WT, AA35-316 and AA1-34 GST-APE1, and the 70 nt-ssDNA structure in vitro.

The online version of this article includes the following source data and figure supplement(s) for figure 2:

**Source data 1.** Raw images of immunoblotting analysis referenced in *Figure 2B*.

**Source data 2.** Raw images of immunoblotting analysis referenced in *Figure 2C*.

**Source data 3.** Raw images of immunoblotting analysis referenced in *Figure 2D*.

**Figure supplement 1.** GST- AP endonuclease 1 (APE1) but not GST associated with beads coupled with single-stranded DNA (ssDNA).

**Figure supplement 1—source data 1.** Raw images of immunoblotting analysis referenced in *Figure 2—figure supplement 1*.

**Figure supplement 2.** Endo/exonuclease activities of WT or various mutant/fragment of AP endonuclease 1 (APE1) in vitro.

*supplement 2C–D*). These observations suggest the importance of the AA1-34 motif of APE1 for its 3'–5' exonuclease activity and the AA35-100 motif within APE1 for its AP endonuclease activity.

## APE1 interacts and recruits ATRIP onto ssDNA in an RPA-independent manner in vitro and promotes the ATR DDR pathway in *Xenopus* egg extracts using a non-catalytic mechanism

We next tested whether and how APE1 might interact with ATRIP directly by protein-protein interaction assays. GST pulldown assays showed that GST-APE1 but not GST directly interacted with His-ATRIP in vitro (*Figure 2A* and *Figure 3A*). Domain dissection experiments revealed that both AA35-316 GST-APE1 and AA1-100 GST-APE1 associated with His-ATRIP to the similar capacity as WT GST-APE1 (*Figure 3*). However, AA101-316 GST-APE1 and other fragments of APE1 tested (i.e. AA1-34, AA35-100, and AA101-200) were deficient for interaction with His-ATRIP (*Figure 3A*). In addition, neither of the point mutants within GST-APE1's active sites (i.e. ED, D306A, and C92A-C98A) affected the APE1-ATRIP interaction (*Figure 3B*), although they are deficient for 3'–5' exonuclease as shown previously (*Lin et al., 2020*). Thus, our findings indicate that AA35-100 within APE1 is required but not sufficient for ATRIP interaction and AA1-100 is the minimum fragment within APE1 sufficient for ATRIP association in vitro (*Figures 2A and 3A*).

Based on the observation of direct APE1-ATRIP interaction (*Figure 3A*), we intended to test whether APE1 could recruit ATRIP onto ssDNA directly in the absence of RPA in vitro. We found that His-ATRIP protein was recruited onto 30 nt- and 80 nt-ssDNA gap structures in the presence of WT GST-APE1 but not GST (compare Lanes 4–6 and Lane 1–3 in 'bead-bound', *Figure 3C*). Due to its deficiency in ssDNA interaction (*Figures 2D and 3A*), AA35-316 GST-APE1 was not recruited to 30 nt- and 80 nt-ssDNA gap structures, which led to the insufficient recruitment of His-ATRIP onto ssDNA (Lanes 10–12 in 'bead-bound', *Figure 3C*). Notably, AA101-316 GST-APE1 was recruited to 30 nt- and 80 nt-ssDNA gap structures but could not recruit ATRIP to ssDNA, due to deficiency in ATRIP association (Lanes 7–9 in 'bead-bound', *Figure 3C*). These observations strongly support that APE1 interacts with ssDNA via its AA1-34 fragment and recruits ATRIP onto ssDNA via its AA1-100 in in vitro reconstitution systems, and that such APE1-mediated ATRIP recruitment onto ssDNA is independent of RPA.

To test the biological significance of APE1-mediated ATRIP onto ssDNA, we performed rescue experiments in APE1-depleted HSS. WT GST-APE1 but not AA101-316 GST-APE1 rescued the recruitment of endogenous ATRIP onto 30 nt- and 80 nt-ssDNA gap structures and subsequent Chk1 phosphorylation, although endogenous RPA70 and RPA32 as well as WT/AA101-316 GST-APE1 associated with ssDNA gap structures in APE1-depleted HSS (compare Lanes 4–6 and Lanes 7–9, *Figure 3D*). This observation indicates the significance of the APE1-ATRIP interaction for ATRIP recruitment onto ssDNA and subsequent ATR DDR pathway activation in *Xenopus* egg extracts. In addition, we performed rescue experiment by adding back AA35-316 GST-APE1 in APE1-depleted HSS system, compared with WT GST-APE1. Our result showed that AA35-316 GST-APE1 was deficient for the interaction with ssDNA gaps and ATRIP recruitment onto ssDNA gaps, and subsequent defective

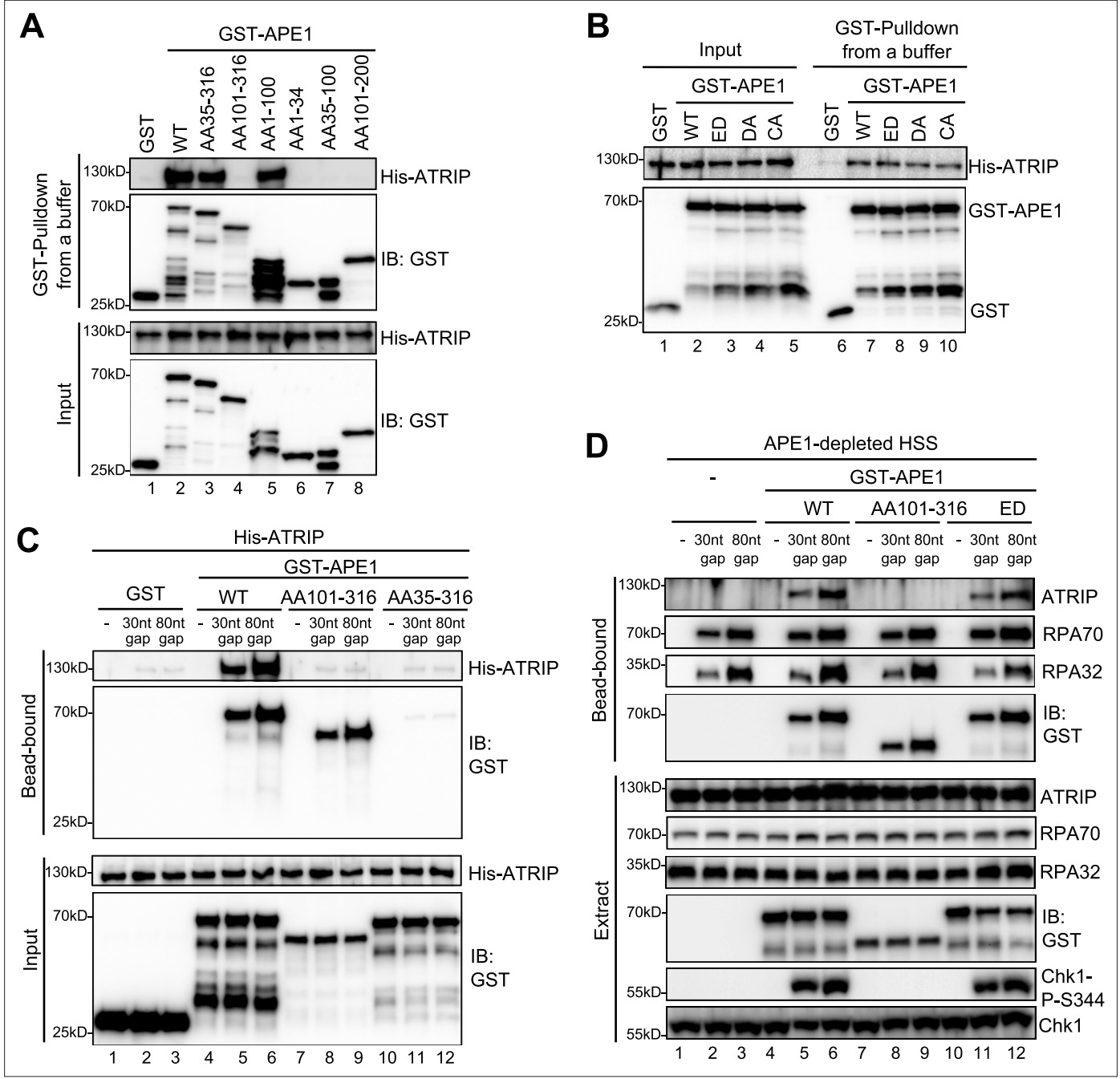

**Figure 3.** AP endonuclease 1 (APE1) interacts and recruits ATRIP onto single-stranded DNA (ssDNA) in an RPA-independent manner in vitro and promotes the ATR DNA damage response (DDR) pathway in *Xenopus* egg extracts using a non-catalytic function mechanism. (**A–B**) GST pulldown assays with GST, WT, or fragment/mutant GST-APE1 as well as His-ATRIP in an interaction buffer. The input and pulldown samples were examined via immunoblotting analysis. (**C**) Streptavidin beads coupled with biotin-labeled double-stranded DNA (dsDNA) with ssDNA gap structures (30 nt or 80 nt) were added to an interaction buffer containing His-ATRIP and GST/GST-tagged proteins (WT, AA101-316, or AA35-316 GST-APE1) as indicated. DNA-bound fractions and input samples were examined via immunoblotting analysis as indicated. (**D**) Streptavidin beads coupled with biotin-labeled dsDNA with ssDNA gap structures (30 nt or 80 nt) were added to APE1-depleted high-speed supernatant (HSS), which was supplemented with GST or GST-tagged proteins (WT, AA101-316, or ED GST-APE1) as indicated. DNA-bound fractions and total extract samples were examined via immunoblotting analysis as indicated.

The online version of this article includes the following source data and figure supplement(s) for figure 3:

**Source data 1.** Raw images of immunoblotting analysis referenced in *Figure 3A*.

**Source data 2.** Raw images of immunoblotting analysis referenced in *Figure 3B*.

*Figure 3 continued on next page*

*Figure 3 continued*

**Source data 3.** Raw images of immunoblotting analysis referenced in *Figure 3C*.

**Source data 4.** Raw images of immunoblotting analysis referenced in *Figure 3D*.

**Figure supplement 1.** Streptavidin beads coupled with biotin-labeled double-stranded DNA (dsDNA) with single-stranded DNA (ssDNA) gap structures (30 nt or 80 nt) were added to AP endonuclease 1 (APE1)-depleted high-speed supernatant (HSS), which was supplemented with WT or AA35-316 GST-APE1 as indicated.

**Figure supplement 1—source data 1.** Raw images of immunoblotting analysis referenced in *Figure 3—figure supplement 1*.

Chk1 phosphorylation in APE1-depleted HSS system, although WT and AA35-316 GST-APE1 were added to APE1-depleted HSS at the similar levels (*Figure 3—figure supplement 1*). Furthermore, ssDNA-coated endogenous RPA protein complex could not recruit AA35-316 GST-APE1 and endogenous ATRIP onto ssDNA in APE1-depleted HSS, although AA35-316 GST-APE1 is proficient for the interaction with ATRIP and RPA protein in vitro (*Figure 3—figure supplement 1*). All these observations indicate that APE1's ssDNA binding capacity and ATRIP interaction are important for both RPA-dependent and RPA-independent ATRIP recruitment onto ssDNA and the ATR/Chk1 in the HSS system.

In light of the significance of APE1 and its 3'–5' exonuclease in the initial end resection of defined SSB structures and subsequent ATR DDR pathway in *Xenopus* HSS system (*Lin et al., 2020*), we sought to test whether APE1's catalytic function plays a vital role in the direct ATRIP regulation and chose to use ED GST-APE1 lacking 3'–5' exonuclease and AP endonuclease (*Figure 2—figure supplement 2A–B*; *Lin et al., 2020*). We found that similar to WT GST-APE1, ED GST-APE1 bound with ssDNA and recruited endogenous ATRIP onto ssDNA and rescued Chk1 phosphorylation in APE1-depleted HSS (Lanes 10–12, *Figure 3D*). This observation suggests that APE1's nuclease activity is dispensable for its direct recruitment of ATRIP onto ssDNA gap structures and subsequent ATR-Chk1 DDR pathway activation in the *Xenopus* HSS system. Together, these findings demonstrate that APE1 directly associates with and recruits ATRIP onto ssDNA in vitro and that APE1 recruits ATRIP onto ssDNA via a non-catalytic function to promote the ATR-Chk1 DDR pathway activation in the *Xenopus* HSS system.

## APE1 interacts with RPA70 and RPA32 via two distinct binding motifs

We have shown that APE1 depletion and RPA depletion led to deficiency or impairment in the recruitment of ATRIP onto ssDNA gaps in *Xenopus* HSS system (*Figure 1B*, *Figure 1—figure supplement 1A*). Next, we sought to determine whether APE1 interacts with RPA directly, and, if so, whether RPA plays a role for APE1-mediated ATRIP recruitment onto ssDNA for ATR DDR in the HSS system. Due to the significant role of RPA in the recruitment of ATR direct activators such as TopBP1 and ETAA1 to ssDNA (*Acevedo et al., 2016*; *Bass et al., 2016*; *Lee et al., 2016*; *Lyu et al., 2019*), it is not technically feasible to test whether RPA depletion directly or indirectly affects the ATR-Chk1 DDR pathway in the HSS system due to potential defective APE1-mediated ATRIP recruitment. Our strategy is to identify RPA-interaction motifs within APE1 in vitro, and to determine whether a mutant APE1 deficient in RPA interaction still recruits ATRIP onto ssDNA in the HSS system.

First, we tested the possibility that recombinant GST-APE1 might interact with purified recombinant His-RPA complex (RPA70, RPA32, and RPA14) by protein-protein interaction assays. Our GST pull-down assays showed that WT GST-APE1 but not GST interacted with His-RPA70 in vitro (*Figure 4A*, *Figure 4—figure supplement 1A*). Almost no noticeable effects were observed for the interaction of ED, DA, and CA GST-APE1 with His-RPA70, suggesting that the E95, D209, D306, C92, and C98 residues in APE1 are not critical for ATRIP interaction (*Figure 4—figure supplement 1B*). Domain dissection experiments revealed that AA35-316 and AA1-100 GST-APE1 associated with His-RPA70 in a similar capacity to WT GST-APE1 (*Figure 4A*). The binding to His-RPA70 was decreased but nevertheless not completely eliminated in other deletion fragments of GST-APE1 tested (i.e. AA101-316, AA1-34, AA35-100, AA101-200) (*Figure 4A*). These observations suggest (i) that the first 100 amino acids of APE1 are important for RPA association and (ii) that more than one binding sites within APE1 may mediate interaction with RPA complex.

Second, to further test our hypothesis of multiple bindings sites of APE1 for its interaction with RPA complex, we performed amino acid sequence alignments of APE1 (*Xenopus* APE1 and human APE1) to several human RPA70-interacting proteins (e.g. ETAA1, ATRIP, RAD9A, NBS1, and Mre11)

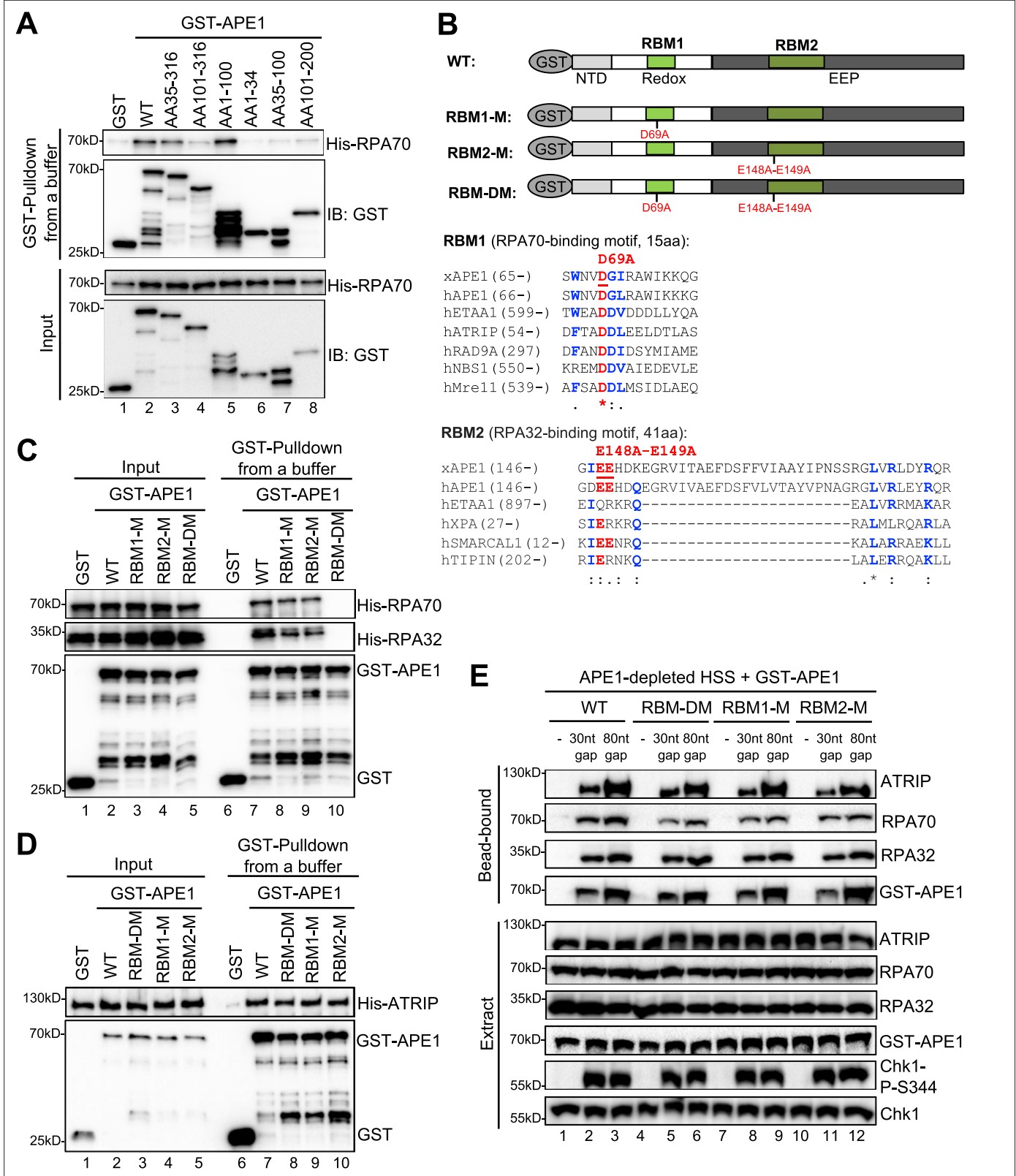

**Figure 4.** AP endonuclease 1 (APE1) interacts with RPA70 and RPA32 via two distinct binding motifs. (**A**) GST pulldown assays with GST, WT, or fragment of GST-APE1 as well as His-RPA in an interaction buffer. The input and pulldown samples were examined via immunoblotting analysis. (**B**) Schematic diagram of APE1 functional domains and its putative RPA-binding motifs (RBM1 and RBM2), as well as sequence alignment of RBM1 and RBM2 from different RPA-interaction proteins. (**C**) GST pulldown assays with GST, WT/mutant GST-APE1, as well as His-RPA protein complex in an interaction buffer.

*Figure 4 continued on next page*

*Figure 4 continued*

The input and pulldown samples were examined via immunoblotting analysis. (**D**) GST pulldown assays with GST, WT/mutant GST-APE1, as well as His-ATRIP protein in an interaction buffer. The input and pulldown samples were examined via immunoblotting analysis. (**E**) Streptavidin beads coupled with biotin-labeled double-stranded DNA (dsDNA) with single-stranded DNA (ssDNA) gap structures (30 nt or 80 nt) were added to APE1-depleted high-speed supernatant (HSS), which was supplemented with WT or RBM mutant GST-APE1 (WT, RBM1-M, RBM2-M or RBM-DM GST-APE1) as indicated. DNA-bound fractions and total extract samples were examined via immunoblotting analysis as indicated.

The online version of this article includes the following source data and figure supplement(s) for figure 4:

**Source data 1.** Raw images of immunoblotting analysis referenced in *Figure 4A*.

**Source data 2.** Raw images of immunoblotting analysis referenced in *Figure 4C*.

**Source data 3.** Raw images of immunoblotting analysis referenced in *Figure 4D*.

**Source data 4.** Raw images of immunoblotting analysis referenced in *Figure 4E*.

**Figure supplement 1.** More characterization of the AP endonuclease 1 (APE1)-RPA interaction.

**Figure supplement 1—source data 1.** Raw images of immunoblotting analysis referenced in *Figure 4—figure supplement 1B*.

**Figure supplement 2.** Endo/exonuclease activities of WT or various mutant/fragment of AP endonuclease 1 (APE1) in vitro.

**Figure supplement 3.** The RPA- AP endonuclease 1 (APE1) interaction promotes APE1 retention on single-stranded DNA (ssDNA).

**Figure supplement 3—source data 1.** Raw images of immunoblotting analysis referenced in *Figure 4—figure supplement 3A*.

**Figure supplement 3—source data 2.** Raw images of immunoblotting analysis referenced in *Figure 4—figure supplement 3C*.

**Figure supplement 3—source data 3.** Raw images of immunoblotting analysis referenced in *Figure 4—figure supplement 3E*.

and RPA32-interacting proteins (e.g. ETAA1, XPA, SMARCAL1, and TIPIN) (*Bass et al., 2016*; *Haahr et al., 2016*; *Lee et al., 2016*), and found that APE1 contains a putative RPA70-binding motif (15AA, designated as RBM1) and a putative RPA32-binding motif (41AA, designated as RBM2) (*Figure 4B*). To determine whether these two possible RPA-binding motifs within APE1 are important for RPA association, we generated single mutant in RBM1 (D69A, designated as RBM1-M) or RBM2 (E148A-E149A, designated as RBM2-M) or in combination (D69A-E148A-E149A, designated as RBM-DM) (*Figure 4B*). GST pulldown assays showed that RBM-DM GST-APE1 was defective for interaction with His-RPA70 and His-RPA32 in vitro, although RPA association was mildly impaired in single mutant RBM1-M and RBM2-M GST-APE1 (*Figure 4C*, *Figure 4—figure supplement 1C*). However, neither of the RPA-binding mutants within APE1 affected its association with His-ATRIP protein (*Figure 4D*). In addition, neither of the RPA-binding mutant GST-APE1 (i.e. RBM1-M, RBM2-M, and RBM-DM) had noticeable effects on APE1's 3'–5' exonuclease and AP endonuclease activity, comparing with WT GST-APE1 under our experimental conditions (*Figure 4—figure supplement 2A–B*). Our observations here suggest that APE1 directly interacts with RPA70 and RPA32 in vitro using two previously uncharacterized distinct motifs within APE1 (i.e. RBM1 and RBM2) (*Figure 4B*).

Our earlier result showed that APE1 depletion led to defective ATRIP recruitment to ssDNA at gap structures and Chk1 phosphorylation in *Xenopus* HSS system (*Figure 1B*). Our rescue experiments showed that similar to WT GST-APE1, the RPA-interaction-deficient mutant GST-APE1 (RBM1-M, RBM2-M, and RBM-DM) rescued the recruitment of endogenous ATRIP and RPA70/RPA32 onto ssDNA and subsequent Chk1 phosphorylation in APE1-depleted HSS (*Figure 4E*). This result suggests that the RPA-APE1 interaction may be dispensable for the APE1-mediated ATRIP recruitment onto ssDNA in *Xenopus* egg extracts. We also tested whether RPA plays any role in the APE1-ssDNA interaction in reconstitution system. Based on the length-dependent APE1 association with ssDNA (*Figure 2C*), we added excess recombinant His-RPA complex and found that APE1 interaction with longer ssDNA (40 nt, 60 nt, and 80 nt) was enhanced by the presence of RPA complex in vitro (*Figure 4—figure supplement 3A–3B*). Similar to WT GST-APE1, RBM-DM GST-APE1 was also recruited to longer ssDNA (40 nt and 80 nt) (*Figure 4—figure supplement 3C–3D*); however, the RPA-stimulated ssDNA interaction of GST-APE1 was impaired when RBM-DM GST-APE1 was compared with WT GST-APE1 (*Figure 4—figure supplement 3C–3D*). These observations suggest that the APE1-RPA interaction may be important for the stabilization of APE1 protein on ssDNA interaction in vitro.

We have shown that both RPA complex and APE1 protein can recruit recombinant ATRIP protein onto ssDNA gaps independently in vitro (*Figures 1C and 3C*), and that APE1 and RPA complex interact with each (*Figure 4*). It is interesting to test whether the RPA-dependent ATRIP recruitment to ssDNA and APE1-dependent ATRIP recruitment to ssDNA is potentially competitive or cooperative. Using

in vitro pulldown assays with beads coupled with 80 nt-ssDNA gap, we found that the presence of both His-RPA protein complex and GST-APE1 protein increased the recruitment of His-ATRIP protein onto ssDNA gaps, compared with each protein individually (Lane 4 vs Lane 2 or 3, *Figure 4—figure supplement 3E–3F*). Interestingly, it seems there is no significance for the recruitment of His-ATRIP protein onto ssDNA when both His-RPA protein complex and GST-APE1 were present, compared with the sum of when individual protein was present (Lane 4 vs [Lane 2+3], *Figure 4—figure supplement 3E–3F*). Our observation suggests that the RPA-mediated and APE1-mediated ATRIP recruitment onto ssDNA is neither cooperative with each other nor competitive/exclusive to each other at least in in vitro reconstitution system under our experimental conditions.

## Discussion

In addition to its critical roles in DNA repair and redox regulation (*Li and Wilson, 2014*; *Tell et al., 2009*), accumulating evidence suggests important roles of APE1 in the activation of the ATR-Chk1 DDR pathway (*Li et al., 2022*; *Lin et al., 2020*; *Vrouwe et al., 2011*). We and others have demonstrated that APE1 and its nuclease activity are important for the ATR-Chk1 DDR activation in response to oxidative DNA damage and ultraviolet damage in mammalian cells (*Li et al., 2022*; *Vrouwe et al., 2011*). Furthermore, APE1 plays an essential role in the initiation step of 3'–5' end resection in SSB-induced ATR-Chk1 DDR pathway via its 3'–5' exonuclease activity in *Xenopus* egg extract system (*Lin et al., 2020*). Mechanistic studies further elucidate that APE1 directly recognizes and binds to SSB site and generate a small ssDNA gap structure via its catalytic function for the subsequent APE2 recruitment and activation for the continuation of SSB end resection (*Lin et al., 2020*). Notably, APE1 forms biomolecular condensates in vitro and in nucleoli independent of its nuclease activity to promote the ATR-Chk1 DDR pathway activation in cancer cells, and APE1 is proposed as a new direct activator of the ATR kinase, in addition to TopBP1 and ETAA1 (*Li and Yan, 2023*; *Li et al., 2022*).

In current study, we have identified and characterized another distinct regulatory mechanism of APE1 in the ATR-Chk1 DDR pathway independent of its nuclease activity. We initially observed that the recruitment of ATRIP onto ssDNA gaps was almost all dependent on APE1 but only partially dependent on RPA in *Xenopus* HSS system (*Figure 1*, *Figure 1—figure supplement 1*). To further elucidate the regulatory mechanism of APE1 in ATRIP recruitment onto ssDNA gaps, we have demonstrated evidence that the APE1 protein interacts with ssDNA in a length-dependent manner and ATRIP protein in vitro, and that WT APE1 protein, but neither the ssDNA-interaction mutant (i.e. AA35-316 APE1) nor the ATRIP-interaction mutant APE1 (i.e. AA101-316 APE1), can recruit recombinant ATRIP protein onto ssDNA gaps in in vitro reconstitution system in the absence of recombinant RPA protein complex, and can recruit endogenous ATRIP protein onto ssDNA gaps for subsequent ATR-Chk1 activation in APE1-depleted *Xenopus* HSS system (*Figures 2–3*). Notably, the nuclease-deficient mutant ED APE1 is still proficient in ATRIP recruitment to the defined ssDNA gaps and subsequent ATR DDR in *Xenopus* egg extracts, similar to WT APE1 (*Figure 3D*). Our data in this study support that APE1 protein directly associates and recruits ATRIP protein onto ssDNA gaps independent of RPA in in vitro reconstitution systems, in addition to the well-established RPA-mediated recruitment of ATRIP onto ssDNA (*Zou and Elledge, 2003*; *Figure 5A*). In the *Xenopus* HSS system in which all RPA-interaction proteins are present (e.g. TopBP1, ETAA1, RAD9A, NBS1, Mre11, TIPIN, XPA, etc.), there are possible two modes for the recruitment of endogenous ATRIP protein onto ssDNA gaps: (Mode #1) APE1-dependent and RPA-dependent; and (Mode #2) APE1-dependent but RPA-independent (*Figure 5B*). In Mode #1: RPA interacts with and recruits ATRIP protein onto RPA-ssDNA in the HSS; however, this RPA-dependent ATRIP recruitment may be inhibited or negatively regulated by a currently uncharacterized Protein X in HSS, and such inhibitory effect by Protein X can be reversed or counteracted by APE1 in the HSS system. In Mode #2, APE1 protein can interact with ATRIP that is not in complex with RPA protein complex, and recruits ATRIP onto ssDNA gaps independent of RPA in HSS. Taken together, our observations in this study suggest a non-catalytic function of APE1 as a direct recruiter of ATRIP protein onto ssDNA gaps in an RPA-dependent and -independent manner for the ATR/Chk1 DDR pathway.

In this study we have characterized a critical role of APE1 as a previously underappreciated ssDNA-interacting protein in the ATR DDR pathway. Our data indicate that APE1 protein directly associates with ssDNA structures (40 nt, 60 nt, and 80 nt) and ssDNA regions (30 nt and 80 nt) in the defined ssDNA gap structures and that the N-terminal 34 amino acids (NT34) within APE1 are important

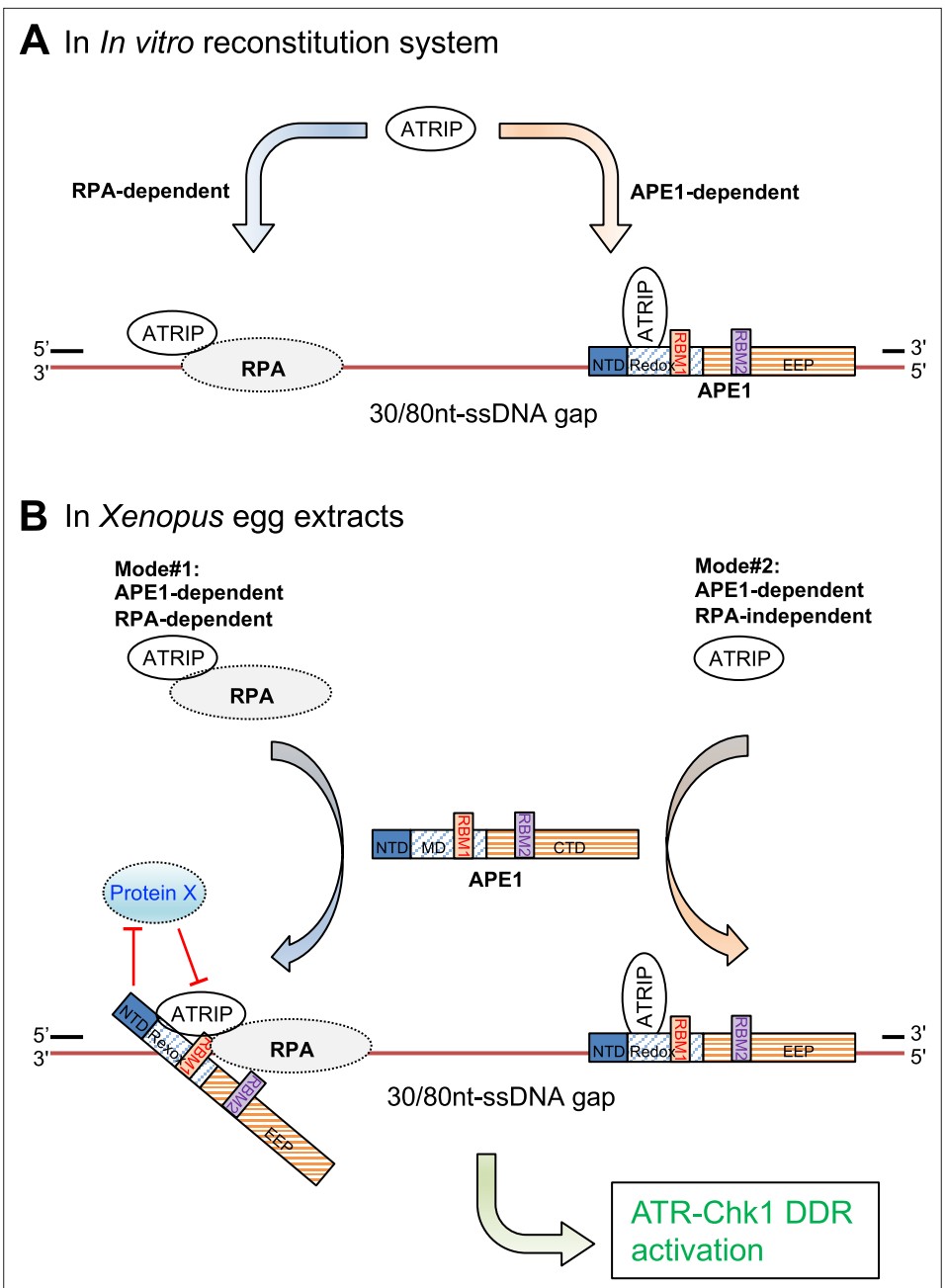

**Figure 5.** A working model of the distinct mechanism of how AP endonuclease 1 (APE1) directly interacts and recruits ATRIP onto single-stranded DNA (ssDNA) independently in vitro and in a concerted fashion in *Xenopus* egg extracts for ATR DNA damage response (DDR) pathway. (**A**) In in vitro reconstitution system, RPA can recruit ATRIP to ssDNA gaps. In parallel, APE1 promotes the recruitment of ATRIP to ssDNA gaps in vitro via APE1 direct interaction with ssDNA and ATRIP protein. The RPA-dependent and APE1-dependent recruitment of ATRIP onto ssDNA is likely independent of each other in in vitro reconstitution system. (**B**) APE1 is required for the recruitment of ATRIP onto ssDNA gaps for the ATR/Chk1 DDR activation in the *Xenopus* high-speed supernatant (HSS) system via two modes: (#1) APE1-dependent and RPA-dependent; and (#2) APE1-dependent but RPA-independent. In Mode #1: RPA interacts with ATRIP protein and recruits ATRIP onto RPA-ssDNA in HSS. However, this Mode #1 may be inhibited by a currently uncharacterized Protein X in HSS, and such inhibitory effect by Protein X can be reversed by APE1. In Mode #2, APE1 can interact with ATRIP that is not in complex with RPA protein complex, and recruits ATRIP onto ssDNA gaps independent of RPA in HSS. Other significant DDR proteins are omitted from this diagram for a simplified illustration. NTD, N-terminal domain; Redox, redox domain; EEP, exonuclease-endonuclease-phosphatase domain; RBM1, RPA70-binding motif; RBM2, RPA32-binding motif.

for and the AA101-316 APE1 is sufficient for such direct ssDNA association (*Figure 2*). Whereas the concentration of *Xenopus* APE1 protein in *Xenopus laevis* egg was estimated to be ~1.5 μM (*Wühr et al., 2014*), the concentration of hAPE1 protein in HEK293T cells was estimated to ~2.8 μM (https://opencell.czbiohub.org/gene/ENSG00000100823) (*Wiśniewski et al., 2014*). Considering most APE1 protein is localized inside of the nucleus of mammalian cells (*Li and Wilson, 2014*), the observed APE1 affinity and interaction with ssDNA in our in vitro EMSA (*Figure 2E*) are physiologically relevant. Notably, the APE1-ssDNA interaction is critical for the recruitment of ATRIP protein onto ssDNA gaps in vitro and in *Xenopus* egg extracts for ATR DDR activation (*Figure 3C*, *Figure 3—figure supplement 1*, *Figure 5*). A previous study has demonstrated that APE1 can incise the AP site within ssDNA in a sequence- and secondary-structure-dependent manner (*Fan et al., 2006*). Although this finding implies that that APE1 may associate with ssDNA, the molecular determinant of APE1 for ssDNA interaction is elusive. A more recent study using EMSA has determined that a 22 nt-ssDNA structure is sufficient for a stable hAPE1-ssDNA complex formation in vitro (*Bazlekowa-Karaban et al., 2019*), which is in line with our estimated ssDNA gap size (~18–26 nt) for APE1 recruitment and ATR DDR activation in response to defined SSB structures in the *Xenopus* HSS system (*Lin et al., 2018*; *Lin et al., 2020*). In addition, a structural biology preprint study has shown that the critical residues in active sites (e.g. Y171, D210, N212, H309) and DNA intercalating residue R177 of hAPE1 are important for the formation and/or stabilization of the hAPE1-ssDNA complex (*Hoitsma et al., 2022*), which is supportive of our observation on the AA101-316 APE1 interaction with ssDNA (*Figure 2D*). Furthermore, mouse APE1 was shown to interact with 20 nt-ssDNA and -dsDNA for exonucleolytic cleavage in vitro (*Liu et al., 2021*). Although it remains unknown how the NT34 motif of APE1 contributes to ssDNA interaction and its 3'–5' exonuclease activity (*Figure 2—figure supplement 2C–D*), this is reminiscent of the regulation of APE2 3'–5' exonuclease by its C-terminal Zf-GRF interaction with ssDNA (*Lin et al., 2018*; *Wallace et al., 2017*). It has been reported that hAPE1 N-terminal domain is important for ssRNA interaction and RNA metabolism as well as biomolecular condensates formation (*Fantini et al., 2010*; *Li et al., 2022*). Future studies are needed to determine how exactly APE1 N-terminal motif and/or EEP domain contributes to ssDNA interaction.

It is an outstanding question in the field of ATR DDR regarding how exactly ATRIP protein is recruited onto ssDNA gaps in an RPA-dependent and/or -independent manner. We have elucidated in this study that the NT100 motif of APE1 is required and sufficient for ATRIP interaction in in vitro protein-protein interaction assays (*Figure 3A*), and that the APE1-ATRIP interaction is essential for the recruitment of recombinant ATRIP protein onto ssDNA gaps in in vitro reconstitution system and endogenous ATRIP protein to ssDNA gaps in *Xenopus* HSS system (*Figure 3C–D*). Our observations support the critical role of the APE1-ATRIP interaction in the APE1-dependent ATRIP recruitment onto ssDNA gaps. For the Mode #1 APE1- and RPA-dependent recruitment of ATRIP onto ssDNA gaps, we hypothesize that an unknown Protein X may negatively regulate ATRIP binding to RPA complex, especially RPA70 N-terminal domain, which has been involved in the recruitment of several DDR proteins to ssDNA such as ETAA1, Mre11, Nbs1, Rad9, p53, and PRIMPOL (*Bhat and Cortez, 2018*). APE1 protein may reverse or counteract with such inhibitory effect of Protein X in the recruitment of ATRIP onto ssDNA. Alternatively, APE1 NT100 motif interacts with ATRIP and RPA70, which may be required for the conformation change and/or stabilization of ATRIP-RPA-ssDNA in the *Xenopus* HSS system (*Figure 5B*). Future studies will test these different scenarios.

To the best of our knowledge, it is the first time showing that APE1 directly associates with RPA via its two distinct motifs (*Figure 4*). A previous study mentioned that His-tagged hAPE1 protein did not associate with purified untagged RPA protein using Ni-NTA-bead-based pulldown assays in vitro (*Fan et al., 2006*). Although we can't explain the discrepancy with our result, we speculate that this may be due to different pulldown methods and optimized experimental conditions. The two RPA-bindings motifs within APE1 bind to RPA70 and RPA32, which is similar to the recently characterzied ETAA-RPA interaction (*Bass et al., 2016*; *Haahr et al., 2016*; *Lee et al., 2016*). Although RPA interaction is needed for the recruitment of ATR activator proteins ETAA1 and TopBP1 to damage sites and ssDNA for ATR activation (*Acevedo et al., 2016*; *Bass et al., 2016*; *Haahr et al., 2016*), our findings suggest that APE1 can associate with ssDNA and interacts and recruits ATRIP onto ssDNA in an RPA-independent manner (*Figures 2–3*). In addition, recent studies using mammalian cells have shown that APE1 and RPA can form biomolecular condensates independently in vitro (*Li et al., 2022*; *Spegg*

*et al., 2023*). Thus, it is interesting to test in the future whether the APE1-RPA interaction may play a role in biomolecular condensate formation on ssDNA gaps.

Taken together, our findings in this study have elucidated distinct mechanisms of how APE1 contributes to the ATR DDR pathway activation via non-catalytic functions.

# Materials and methods

**Key resources table**

| Reagent type (species) or resource | Designation | Source or reference | Identifiers | Additional information |
|---|---|---|---|---|
| Cell line (*Homo sapiens*) | U2OS | ATCC | HTB-96 | NA |
| Strain, strain background (*Escherichia coli*) | BL21(DE3) | Sigma-Aldrich | CMC0015 | Electrocompetent cells |
| Transfected construct (human) | siRNA to APEX1 (ON-TARGETplus SMARTpool) | Dharmacon/Horizon Discovery Lts. and *Li et al., 2022* | L-010237-00-0005 | Transfected construct (human) |
| Sequence-based reagent (oligonucleotides) | 70 nt FAM-ssDNA structure | This study and *Lin et al., 2020* | Oligo#1 | FAM-5'-TCGGTACCCGGGGATCCTCTAGAGTCGACCTGCAGGCATGCAAGCTTGGCGTAATCATGGTCATAGCTGT-3' |
| Sequence-based reagent (oligonucleotides) | 60 nt Biotin-labeled top strand | This study | Oligo#2 | Biotin-5'-GGGTAACGCCAGGGTTTTCCCAGTCACGACGTTGTAAAACGACGGCCAGTGAATTCGAGC-3' |
| Sequence-based reagent (oligonucleotides) | 10 nt top strand | This study | Oligo#3 | 5'-TGCAGGCATG-3' |
| Sequence-based reagent (oligonucleotides) | 100 nt bottom strand | This study | Oligo#4 | 5'-CATGCCTGCAGGTCGACTCTAGAGGATCCCCGGGTACCGAGCTCGAATTCACTGGCCGTCGTTTTACAACGTCGTGACTGGGAAAACCCTGGCGTTACCC- 3' |
| Sequence-based reagent (oligonucleotides) | 10 nt Biotin-labeled top strand | This study | Oligo#5 | Biotin-5'-GGGTAACGCC-3' |
| Sequence-based reagent (oligonucleotides) | 70 nt Biotin-ssDNA structure | This study | Oligo#6 | Bioin-5'- ACAGCTATGACCATGATTACGCCAAGCTTGCATGCCTGCAGGTCGACTCTAGAGGATCCCCGGGTACCGA-3' |
| Sequence-based reagent (oligonucleotides) | 10 nt Biotin-ssDNA structure | This study and *Ha et al., 2020* | Oligo#7 | Bioin-5'-GGTCGACTCT-3' |
| Sequence-based reagent (oligonucleotides) | 20nt Biotin-ssDNA structure | This study and *Ha et al., 2020* | Oligo#8 | Bioin-5'- GGTCGACTCTAGAGGATCCC-3' |
| Sequence-based reagent (oligonucleotides) | 40 nt Biotin-ssDNA structure | This study and *Ha et al., 2020* | Oligo#9 | Bioin-5'- GGTCGACTCTAGAGGATCCCCGGGTACCGAGCTCGAATTC-3' |
| Sequence-based reagent (oligonucleotides) | 60 nt Biotin-ssDNA structure | This study and *Ha et al., 2020* | Oligo#10 | Bioin-5'-GGTCGACTCTAGAGGATCCCCGGGTACCGAGCTCGAATTCACTGGCCGTCGTTTTACAAC-3' |
| Sequence-based reagent (oligonucleotides) | 80 nt Biotin-ssDNA structure | This study | Oligo#11 | Bioin-5'- GGTCGACTCTAGAGGATCCCCGGGTACCGAGCTCGAATTCACTGGCCGTCGTTTTACAACGTCGTGACTGGGAAAACCCT-3' |
| Antibody | Anti-*Xenopus* APE1 (Rabbit polyclonal) | *Lin et al., 2020* | | IB (1:2000) |
| Antibody | Anti-*Xenopus* ATRIP (Rabbit polyclonal) | *Willis et al., 2013* | | IB (1:2000) |

*Continued on next page*

*Continued*

| Reagent type (species) or resource | Designation | Source or reference | Identifiers | Additional information |
|---|---|---|---|---|
| Antibody | Anti-*Xenopus* RPA70 (Rabbit polyclonal) | *Acevedo et al., 2016* | | IB (1:5000) |
| Antibody | Anti-*Xenopus* RPA32 (Rabbit polyclonal) | *Acevedo et al., 2016* | | IB (1:5000) |
| Antibody | Anti-Chk1-P-S345 (Rabbit monoclonal) | Cell Signaling Technology | Cat#2348 | IB (1:2000) |
| Antibody | Anti-Chk1-P-S317 (Rabbit monoclonal) | Cell Signaling Technology | Cat#12302 | IB (1:1000) |
| Antibody | Anti-Chk1 (Mouse monoclonal) | Santa Cruz Biotechnology | Cat#sc-8408 | IB (1:2000) |
| Antibody | Anti-GST (Mouse monoclonal) | Santa Cruz Biotechnology | Cat#sc-138 | IB (1:5000) |
| Antibody | Anti-His (Mouse monoclonal) | Santa Cruz Biotechnology | Cat#sc-8036 | IB (1:1000) |
| Antibody | Anti-human APE1 (Mouse monoclonal) | Santa Cruz Biotechnology | Cat#sc-17774 | IB (1:2000) |
| Antibody | Anti-PCNA (Mouse monoclonal) | Santa Cruz Biotechnology | Cat#sc-56 | IB (1:4000) |

## Experimental procedures for *Xenopus* egg extracts, DDR signaling technology, and plasmid DNA bound fraction isolation in *Xenopus* egg extracts

The preparation of *Xenopus* HSS and immunodepletion of target proteins in HSS were described previously (*Cupello et al., 2019*; *Cupello et al., 2016*; *Lebofsky et al., 2009*; *Lin et al., 2020*; *Willis et al., 2012*). For the DDR signaling experiments, typically different plasmid DNA was mixed with HSS to final concentrations (e.g. 75 ng/μL) for a 45 min incubation at room temperature using a protocol similar to previously described (*Lin et al., 2019*). Reaction mixture was added with sample buffer followed by examination via immunoblotting analysis. For DNA-bound protein isolation from HSS system, a detailed method has been described previously (*Lin et al., 2018*). Briefly, reaction mixture was diluted with egg lysis buffer followed by spinning through a sucrose cushion at 10,000 rpm at 4°C for 15 min. After aspiration, the DNA-bound protein factions were analyzed via immunoblotting analysis. *Xenopus* use is approved by UNCC IACUC (#22-023 and #19-004).

## Cell culture, and knockdown of APE1 and preparation of cell lysates

Human osteosarcoma cell line U2OS cells were purchased from and authenticated by American Type Culture Collection (ATCC, Cat#HTB-96), and were tested negative for mycoplasma contamination. U2OS cells were cultured in DMEM with 10% FBS and penicillin (100 U/mL) and streptomycin (100 μg/mL) at 37°C in $CO_2$ incubator (5%). For siRNA-mediated APE1-KD experiment, siRNA On-Targetplus SMARTpool for APE1 was transfected to U2OS cells (30% confluence) using Lipofectamine RNAiMAX reagent method, as previously described (*Li et al., 2022*). For oxidative stress experiment, U2OS cells were treated with $H_2O_2$ (1.25 mM) for 2 hr before cell collection and the total cell lysate preparation for immunoblotting analysis, as recently described (*Li et al., 2022*).

## Preparation of various plasmids and FAM/biotin-labeled DNA structures

The preparation of control (CTL) plasmid and SSB plasmid was described previously (*Lin et al., 2018*; *Lin et al., 2019*). To generate the gap plasmid structure ('Gap' plasmid), the SSB plasmid was treated

by WT-GST-APE1 protein in a reaction buffer (50 mM HEPES pH 7.4, 60 mM NaCl, 2 mM MgCl$_2$, and 2 mM DTT) followed by phenol-chloroform extraction and purification using a similar procedure described previously (*Lin et al., 2020*).

The 39 bp FAM-labeled dsDNA-AP structure for APE1 endonuclease assays was prepared as previously described (*Lin et al., 2020*). As shown previously (*Lin et al., 2020*), the 70 bp FAM-dsDNA structure was prepared and treated with Nt.BstNBI and CIP to make the FAM-dsDNA-SSB for APE1 exonuclease assays. The FAM-dsDNA-SSB structure was purified from agarose via QIAquick gel extraction and then phenol-chloroform extraction. The 70 nt FAM-ssDNA structure in *Figure 2E* was synthesize as Oligo#1.

The 100 bp biotin-dsDNA structure with a 30 nt- or 80 nt-ssDNA gap (30 nt gap or 80 nt gap) in the middle was created by annealing of three complementary oligos in same molar ratio at 95–100°C for 5 min followed by natural cooling down at room temperature for ~30 min. For the 30 nt-ssDNA gap, the three complementary oligos are 60 nt biotin-labeled top strand Oligo#2, 10 nt top stand Oligo#3, and 100 nt bottom strand Oligo#4. For the 80 nt-ssDNA gap, the three complementary oligos are 10 nt biotin-labeled top strand Oligo#5, 10 nt top stand Oligo#3, and 100 nt bottom strand Oligo#4. The 70 nt biotin-ssDNA structure in *Figure 2D* and *Figure 2—figure supplement 1* was designed as Oligo#6. The biotin-ssDNA structures with different lengths of ssDNA were synthesized as Oligo#7 (10 nt), Oligo#8 (20 nt), Oligo#9 (40 nt), Oligo#10 (60nt), and Oligo#11 (80 nt) and most of them were described previously (*Ha et al., 2020*).

## Recombinant DNA and proteins, and immunoblotting analysis

The preparation of recombinant WT, mutants (ED, DA, CA), and some fragments (AA35-316, AA101-316, AA101-200) of pGEX-4T1-APE1 was described previously (*Lin et al., 2020*). Other fragments of GST-APE1 (e.g. AA1-100, AA1-34, AA35-100) were generated by PCR of respective fragment and subcloned into pGEX-4T1. RBM1-M, RBM2-M, and RBM-DM pGEX-4T1-APE1 were mutated with QuikChange IIXL Site-Directed Mutagenesis kit (Agilent) and purified by QIAprep spin miniprep kit. His-RPA trimer expression plasmid was described previously (*Acevedo et al., 2016*). His-ATRIP expression plasmid was generated by PCR full-length *Xenopus* ATRIP into pET28A at BamHI and XhoI sites. The induction/expression, purification, and validation of GST or His-tagged recombinant proteins from BL21(DE3) *Escherichia coli* cells (VWR Cat#80030-326) were performed following vendor's standard protocol. Immunoblotting analysis was performed following similar methods described previously (*Lin et al., 2020*; *Yan and Michael, 2009*).

## GST pulldown assays and DNA binding assays

The GST-pulldown experiments were performed in an interaction buffer using similar methods as described previously (*Lin et al., 2018*; *Lin et al., 2020*). Methods for the DNA binding assays have been described previously (*Lin et al., 2018*; *Lin et al., 2020*). Briefly, streptavidin Dynabeads coupled with various biotin-labeled structures (e.g. biotin-ssDNA, biotin-dsDNA, or biotin-dsDNA with ssDNA gap) was incubated with various recombinant proteins in a buffer (80 mM NaCl, 20 mM glycerophosphate, 2.5 mM EGTA, 0.01% NP-40, 10 mM MgCl$_2$, 100 μg/mL BSA, 10 mM DTT, and 10 mM HEPES-KOH, pH 7.5) or with the HSS as indicated. After washing, the bead-bound fractions and input samples were examined via immunoblotting analysis.

## In vitro endo/exonuclease assays

For in vitro APE1 endo/exonuclease assays, the FAM-dsDNA-AP structure or FAM-dsDNA-SSB structure was treated with WT, mutant or fragment of GST-APE1 in APE1 reaction buffer at 37°C, as described previously (*Lin et al., 2020*). The reactions were quenched with TBE-urea sample buffer and denatured for 5 min at 95°C. The samples were examined on TBE-urea PAGE gel and imaged with a Bio-Rad imager.

## Electrophoretic mobility shift assays

The EMSA for testing DNA-protein interaction were similar to methods described previously (*Lin et al., 2020*). Briefly, an increasing concentration of proteins were incubated with 10 nM of FAM-labeled DNA structures in EMSA reaction buffer. Reactions were examined on a TBE native gel and imaged by a Bio-Rad imager.

## Quantification, statistical analysis, and reproducibility

The data presented are representative of three biological replicates unless otherwise specified. All statistical analysis of the intensity of protein of interest was performed between individual samples using GraphPad Prism with unpaired $t$-test (*Figure 4—figure supplement 3B*) or paired $t$-test (all other analyses). Results are shown as mean ± standard deviation (SD) for three independent experiments (p values are as indicated, n=3). $p < 0.05$ is considered as significantly different.

## Material availability statement

Materials generated in this study can be accessed by contacting the corresponding author.

## Acknowledgements

We thank Drs. Matthew Michael, Karlene Cimprich, and Howard Lindsay for reagents. The Yan lab was supported, in part, by grants from the NIH/NCI (R01CA225637 and R03CA270663) and the NIH/NIEHS (R21ES032966), and funds from UNC Charlotte.

## Additional information

### Funding

| Funder | Grant reference number | Author |
|---|---|---|
| National Institutes of Health | R01CA225637 | Shan Yan |
| National Institutes of Health | R03CA270663 | Shan Yan |
| National Institutes of Health | R21ES032966 | Shan Yan |
| University of North Carolina at Charlotte | Seed grants | Shan Yan |

The funders had no role in study design, data collection and interpretation, or the decision to submit the work for publication.

### Author contributions

Yunfeng Lin, Conceptualization, Data curation, Investigation, Methodology, Writing - original draft, Writing - review and editing; Jia Li, Haichao Zhao, Kelly McGhee, Data curation, Investigation; Anne McMahon, Data curation, Investigation, Writing - review and editing; Shan Yan, Conceptualization, Data curation, Supervision, Funding acquisition, Investigation, Methodology, Writing - original draft, Writing - review and editing

### Author ORCIDs

Shan Yan ⓘD http://orcid.org/0000-0001-8106-6295

### Ethics

The care and use of X. laevis was approved by the Institutional Animal Care and Use Committee (IACUC) at University of North Carolina at Charlotte (IACUC-22-023 and IACUC-19-004).

### Decision letter and Author response

Decision letter https://doi.org/10.7554/eLife.82324.sa1
Author response https://doi.org/10.7554/eLife.82324.sa2

## Additional files

### Supplementary files
• MDAR checklist

## Data availability

All data generated or analysed during this study are included in the manuscript and supporting files.

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
