## [Editor Report]

This important paper provides new insight into the mechanism of the activation of DNA damage checkpoint (DDR) in response to the single-stranded DNAs (ssDNAs). The authors used *Xenopus* egg extracts and a reconstitution reaction with purified proteins and presented convincing results to support the authors' claims on a non-catalytic role of APE1 endonuclease to recruit DDR activator, ATRIP, to the ssDNA for DDR activation. The work would be of interest to researchers who work on the cell cycle and DNA damage responses as well as DNA repair.

---

## [Decision Letter]

**Decision letter after peer review:**

Thank you for submitting your article "APE1 recruits ATRIP to ssDNA in an RPA-independent manner to promote the ATR DNA damage response" for consideration by *eLife*. Your article has been reviewed by 3 peer reviewers, including Akira Shinohara as the Reviewing Editor and Reviewer #1, and the evaluation has been overseen by Kevin Struhl as the Senior Editor.

In addition to the Essential Revisions, please address also the other reviewers' comments. If you think the experiments suggested are not reasonable, please provide relevant arguments.

Essential revisions:

1. It is very important to re-evaluate the role of RPA in the ATRIP recruitment to ssDNAs in in vitro *Xenopus* high-speed supernatant (HSS) by depleting RPA (and both RPA and APE1, if still ATRIP binding in the absence of RPA) from the HSS. In this experiment, the authors would be able to nicely show RPA-independent (APE1-dependent) ATRIP binding to the gapped DNAs.

2. To evaluate the role of APE1 in the ATR signaling in vivo, the authors need to analyze Chk1 phosphorylation or other ATR activation marker(s) in APE1-siRNA depletion experiments in a cell line (or a deletion mutant of yeast if the mechanism is conserved) with exposures to chemicals which induce gapped ssDNAs such as H2O2.

3. It is essential to quantify all biochemical results with proper statistics and to mention the reproducibility for all in vitro and future in vivo experiments.

*Reviewer #1 (Recommendations for the authors):*

The paper needs some additional work to validate the main conclusion and should show the quantification of all biochemical results.

1. Figures 1A, B and Figure 4D clearly showed that APE1-depletion from the extract does not induce ATRIP-binding to ssDNA substrates and, as a result, impairs Chk1-phosphorylation even in the presence of RPA on the ssDNAs. This strongly suggests that RPA-binding to ssDNAs is NOT sufficient for ATRIP-recruitment (and ATR activation) on the ssDNAs at least in the *Xenopus* HSS system, implying, a new role of RPA in ATR-activation other than the simple recruitment of ATRIP; the cooperation with APE1. A simple idea is that RPA directly recruits APE1, and then APE1 recruits ATRIP "without" direct interaction with ssDNA, rather a model proposed in Figure 5B with a negative regulator "X", which partly comes from complicated regulation of APE1 binding to the DNAs with a negative regulation by 35-100 aa region of the protein. To approve or deney the possibility and further elucidate the role of DNA binding of APE1 in the checkpoint activation, it is critical to test the role of APE1 DNA binding in ATR/CHK1 signaling by examining AA35-316 for Chk1 activation in APE-depleted *Xenopus* extracts as shown in Figure 3D.

2. It is important to check the binding of ATRIP to ssDNA in the presence of both APE1 and RPA by checking the effect of differential ratios of APE1 and RPA on in vitro binding of ATRIP to the ssDNA as shown in Figures 1C and 3C. The authors can address whether the abilities of the two proteins to recruit ATRIP to the ssDNA is competitive or cooperative or independent.

3. RBM1 in APE1 shows sequence similarity to an N-terminal region of ATRIP for RAP70N (OB-F of RPA70) binding. If so, the RBM1 of APE1 and the ATRIP region should show competitive bindings to RPA70N. This should be examined by competitive pull-down assay using purified APE1, ATRIP-N-terminal region, and RPA70N.

4. Please demonstrate the in vivo role of APE1(APEX1) in ATR activation using siRNA inhibition in mammalian cells or APE1 deletion of yeast cells.

5. Please show quantification of all blots with error bars with proper statistics (or claim on the reproducibility of the results)

*Reviewer #2 (Recommendations for the authors):*

This reviewer suggests that the authors consider the following points to validate the model and increase the impact.

1. The authors need to test the effect of RPA depletion (Figure 1, Figure 3, and Figure 4, at least one experiment) because DNA binding of ATRIP and ATR activation was always associated with RPA-DNA binding.

2. It is not clear how PCNA is equally loaded on the gapped and non-gapped DNA substrates (Figure 1).

3. The proposed model suggests that RPA prevents APE1 from ssDNA recognition (Figure 5). The significance of APE1 in ATR recruitment needs to be further discussed or experimentally addressed. RPA interacts weakly with short stretches of ssDNA. Does APE1 act specifically on a gapped DNA to stimulate the ATR pathway? Kim et al. (2005) showed that annealing of dA/dT oligos also triggers the ATR pathway. They could test whether APE1 also acts on annealed oligos. APE1 is an endonuclease that cleaves the DNA phosphodiester backbone at AP sites, leaving a 1 nt gap. Does APE1 activate the ATR pathway by DNA circles containing AP sites (Figure 1)?

*Reviewer #3 (Recommendations for the authors):*

1. The title is misleading. As depicted in the model in Figure 5B, the main role of APE1 in promoting ATR activation in HSS extracts seems to be to counteract a putative inhibitor of ATRIP recruitment. However, there is no data in the paper showing that this role is linked to the RPA-independent mechanism of ATRIP recruitment. It is suggested to remove "RPA-independent manner" from the title.

2. An important experiment missing is to test the effect of RPA on APE1-dependent ATRIP recruitment. In the in vitro experimental setting, the authors should compare absence versus presence of RPA to address if the addition of RPA strongly enhances ATRIP recruitment when APE1 is present.

3. The authors never discuss the importance of APE1 for ATR activation in different contexts and different lesion types. Is the role of APE1 in promoting ATRIP recruitment broadly required for ATR activation? If so, one would predict that cells lacking APE1 should exhibit deficient ATR signaling in response to several types of genotoxins (and is that the case?). If not, then how do the authors explain the central importance of APE1 in ATRIP recruitment in their system given that ssDNA is a general trigger of ATR signaling?

4. In Figures 1A and 1B, APE1 depletion completely blocked ATRIP recruitment to Gap plasmids and dsDNA containing 30nt and 80nt ssDNA gaps. These results indicated that APE1 is essential for the recruitment of ATRIP onto RPA-coated ssDNA in *Xenopus* HSS and RPA cannot bind ATRIP without the presence of APE1. However, this explanation conflicts with the proposed model in Figure 5A, in which RPA is able to recruit ATRIP without APE1. As such, the working model in Figure 5A is not supported by the HSS experiments performed in Figure 1. The authors need to clarify this point of confusion, which is probably attributed to the model in Figure 5A being based only on in vitro data.

5. In Figure 1D the authors claim that APE1 increases the recruitment of ATRIP, however, such an increase in recruitment is not obvious from the presented blot, and careful quantitation (with replicates and statistics) should be presented to support the claim.

6. In Figures 2B and 2C, the authors show that APE1-ssDNA association occurs in a length-dependent manner in vitro, where no ssDNA shorter than 30nt is shown to interact with APE1. However, this is in conflict with Figure 1A, in which Gap plasmid resected to reveal 18-26nt of ssDNA (refer to line 129) is still able to robustly interact with APE1. The authors should address this discrepancy.

7. The concentrations of APE1 proteins used for EMSA assay should be disclosed to show the binding affinity of APE1 and ssDNA. Meanwhile, whether this affinity is physiologically relevant?

8. In Figure 4A, the pulldown of GST-APE1 AA1-34 still pulls down RPA70 even though both RPA-binding motifs are entirely absent. This conflicts with the result seen in the double RBM mutant, where no RPA is pulled down at all in Figure 4C. The authors should address this discrepancy.

9. Overall, writing should be improved throughout the manuscript.

---

## [Author Response]

Essential revisions:1. It is very important to re-evaluate the role of RPA in the ATRIP recruitment to ssDNAs in in vitro *Xenopus* high-speed supernatant (HSS) by depleting RPA (and both RPA and APE1, if still ATRIP binding in the absence of RPA) from the HSS. In this experiment, the authors would be able to nicely show RPA-independent (APE1-dependent) ATRIP binding to the gapped DNAs.

Thanks for the suggestion. We performed RPA depletion in HSS and added additional paragraph in the revised manuscript to describe the new data (Page 7 Line 165-184):

“To re-evaluate the role of RPA in the recruitment of ATRIP onto ssDNA gaps, we performed RPA depletion experiment in HSS and characterized the phenotype of RPA depletion in ATRIP recruitment and ATR-Chk1 DDR pathway. With anti-RPA antibodies, majority of the endogenous RPA protein complex, if not all, was immunodepleted from HSS (Extract panel, Figure 1—figure supplement 1A), and such RPA depletion led to almost no binding of RPA protein complex (RPA70 and RPA32) onto the defined ssDNA gaps and impairment of the ssDNA-induced Chk1 phosphorylation (Figure 1—figure supplement 1A). Whereas RPA deletion did not decrease the recruitment of APE1 onto ssDNA gaps, the recruitment of endogenous ATRIP protein onto ssDNA gaps (30nt and 80nt) was impaired in RPA-depleted HSS (Bead-bound panel and quantification panel, Figure 1—figure supplement 1A). We noted that anti-RPA antibodies co-depleted some endogenous ATRIP protein from HSS, but had almost no co-depletion of endogenous APE1 protein in HSS (Lanes 1-3 vs Lanes 4-6 in Extract panel, Figure 1—figure supplement 1A). This co-depletion of endogenous ATRIP protein is similar to a previous observation by the Dunphy group (Kim et al., 2005), and may suggest a tight complex formation between endogenous ATRIP and RPA protein complex in *Xenopus* egg extracts. Our quantifications of ssDNA-bound ATRIP normalized to available ATRIP protein in HSS showed that RPA depletion almost had no effect on ATRIP’s binding to the 30nt-ssDNA gap, and only mildly decreased the binding of ATRIP (~20%) to 80nt-ssDNA gap (quantification panel, *Figure 1—figure supplement 1A*). These observations suggest that the recruitment of ATRIP onto ssDNA gaps is RPA-dependent and RPA-independent in the HSS system.”

Our finding in Figure 1B have shown that APE1 is required for almost all ATRIP recruitment onto ssDNA gaps in HSS system. This observation is further supported our quantification of ATRIP recruitment to ssDNA gaps normalized to total ATRIP in extract from three independent repeated experiments (Lane 2 vs Lane 5, Lane 3 vs Lane 6, quantification panel, *Figure 1B*). The combined data from Figure 1B and Figure 1—figure supplement 1A suggest two modes of ATRIP recruitment onto ssDNA gaps in *Xenopus* HSS: (#1) APE1-dependent and RPA-dependent; and (#2) APE1-dependent but RPA-independent, which is illustrated in revised *Figure 5B*. In mode #1: RPA interacts with ATRIP protein and recruits ATRIP onto RPA-ssDNA in HSS; however, this mode #1 may be inhibited by a currently uncharacterized Protein X in HSS, and such inhibitory effect by Protein X can be reversed by APE1. In Mode #2, APE1 can interact with ATRIP that is not in complex with RPA protein complex, and recruits ATRIP onto ssDNA gaps independent of RPA in HSS.

2. To evaluate the role of APE1 in the ATR signaling in vivo, the authors need to analyze Chk1 phosphorylation or other ATR activation marker(s) in APE1-siRNA depletion experiments in a cell line (or a deletion mutant of yeast if the mechanism is conserved) with exposures to chemicals which induce gapped ssDNAs such as H2O2.

We thank the suggestion. In our recently published paper (*Nucleic Acids Res*, 2022, 50: 10503-10525, PMID: 36200829), we have demonstrated evidence showing that siRNA-mediated knockdown of endogenous APE1 significantly reduced Chk1 phosphorylation at Ser345 in human breast cancer cell MDA-MB-231 and human pancreatic cancer cell PANC1 cells. Another prior study also shows the role of APE1 in ATR DDR activation in response to UV damage in nucleotide excision repair-deficient cells (J Cell Sci, 2011, 124: 435-446, PMID: 21224401). In this revised manuscript for *eLife*, we provide additional evidence using human osteosarcoma cells U2OS cells and show that siRNA-mediated knockdown of human APE1 impaired the H2O2-induced human Chk1 phosphorylation at Ser345 and Ser317. We added a new paragraph describing this new data in our revised manuscript (Page 7-8, Line 185-196):

“Because ssDNA gaps have been widely accepted as a central platform for protein recruitment and activation of the ATR/Chk1 DDR pathway (Cimprich and Cortez, 2008; Marechal and Zou, 2015), we are prompted to evaluate the role of APE1 in the ATR/Chk1 DDR pathway in cultured cells. Hydrogen peroxide-induced oxidative DNA damage triggered Chk1 phosphorylation at Ser345 and Ser317 in human osteosarcoma U2OS cells, suggesting the activation of ATR DDR (Figure 1—figure supplement 1B). Notably, siRNA-mediated APE1 knockdown of human APE1 in U2OS cells compromised the hydrogen peroxide-induced Chk1 phosphorylation, suggesting that APE1 is important for the ATR/Chk1 DDR in cultured human cells (Figure 1—figure supplement 1B). Our observation here is consistent with two prior studies showing that APE1 is important for the ATR DDR pathway activation in response to oxidative DNA damage in human cancer cells MDA-MB-231 and PANC1 cells, and UV-damage in non-dividing nucleotide excision repair-deficient (i.e., XPC^-/-^) cells (Li et al., 2022; Vrouwe et al., 2011).”

3. It is essential to quantify all biochemical results with proper statistics and to mention the reproducibility for all in vitro and future in vivo experiments.

Thanks for the suggestion. For all biochemical immunoblotting analysis in the revised manuscript, we have at least three independent experimentations showing similar observations. The two more repeated experiments other than those assembled into the final figures and supplementary figures are included in the uploaded source data. Although our response to this question is most time-consuming, we now have all reproducible data collected and included in our revised manuscript.

It seems too overwhelming and sort of demanding to quantify and perform statistical analysis of all bands from immunoblotting analysis results. However, we did perform quantification and statistical analysis in following figures/supplementary figures to support our major claims in the revised manuscript:

1)ATRIP in Bead/Extract in Figure 1A and Figure 1B;2)ATRIP in Bead/Input in Figure 1D;3)APE1 and ATRIP in Bead/Extract in Figure 1—figure supplement 1A;4)hChk1-P-Ser345/Chk1 in Figure 1—figure supplement 1B;5)ATRIP in Bead/Input in Figure 1—figure supplement 1D;6)His-RPA70 PD/Input in Figure 4—figure supplement 1A;7)His-RPA70 and His-RPA32 PD/Input in Figure 4—figure supplement 1C;8)GST-APE1 Bead/Input in Figure 4—figure supplement 3B;9)GST-APE1 Bead/Input in Figure 4—figure supplement 3D;10)ATRIP DNA/Input in Figure 4—figure supplement 3F;

In addition, we added a section of quantification and statistical analysis in the Materials and method section of our revision (Page 22, Line 566-572):

“Quantification, statistical analysis and reproducibility

The data presented are representative of three biological replicates unless otherwise specified. All statistical analysis of the intensity of protein of interest was performed between individual samples using GraphPad Prism with unpaired *t*-test (Figure 4—figure supplement 3B) or paired *t*-test (all other analyses). Results are shown as mean ± standard deviation (SD) for three independent experiments (p values are as indicated, n=3). p<0.05 is considered as significantly different.”

Reviewer #1 (Recommendations for the authors):The paper needs some additional work to validate the main conclusion and should show the quantification of all biochemical results.1. Figures 1A, B and Figure 4D clearly showed that APE1-depletion from the extract does not induce ATRIP-binding to ssDNA substrates and, as a result, impairs Chk1-phosphorylation even in the presence of RPA on the ssDNAs. This strongly suggests that RPA-binding to ssDNAs is NOT sufficient for ATRIP-recruitment (and ATR activation) on the ssDNAs at least in the Xenopus HSS system, implying, a new role of RPA in ATR-activation other than the simple recruitment of ATRIP; the cooperation with APE1. A simple idea is that RPA directly recruits APE1, and then APE1 recruits ATRIP "without" direct interaction with ssDNA, rather a model proposed in Figure 5B with a negative regulator "X", which partly comes from complicated regulation of APE1 binding to the DNAs with a negative regulation by 35-100 aa region of the protein. To approve or deney the possibility and further elucidate the role of DNA binding of APE1 in the checkpoint activation, it is critical to test the role of APE1 DNA binding in ATR/CHK1 signaling by examining AA35-316 for Chk1 activation in APE-depleted *Xenopus* extracts as shown in Figure 3D.

We appreciate the reasoning and suggestion from this Reviewer. Following the suggestion, we performed the rescue experiment in APE1-depleted HSS, and the below description of the result (Page 10-11, Line 281-291):

“In addition, we performed rescue experiment by adding back AA35-316 GST-APE1 in APE1-depleted HSS system, compared with WT GST-APE1. Our result showed that AA35-316 GST-APE1 was deficient for the interaction with ssDNA gaps and ATRIP recruitment onto ssDNA gaps, and subsequent defective Chk1 phosphorylation in APE1-depleted HSS system, although WT and AA35-316 GST-APE1 were added to APE1-deplted HSS at the similar levels (Figure 3—figure supplement 1). Furthermore, ssDNA-coated endogenous RPA protein complex could not recruit AA35-316 GST-APE1 and endogenous ATRIP onto ssDNA in APE1-depleted HSS, although AA35-316 GST-APE1 is proficient for the interaction with ATRIP and RPA protein in vitro (Figure 3—figure supplement 1). All these observations indicate that APE1’s ssDNA binding capacity and ATRIP interaction are important for both RPA-dependent and RPA-independent ATRIP recruitment onto ssDNA and the ATR/Chk1 in the HSS system.”

These new data seem do not support the sequential RPA-APE1-ATRIP recruitment model suggested by the reviewer. Our in vitro reconstitution systems in the absence of RPA protein complex have shown that recombinant His-ATRIP protein was not recruited onto ssDNA gaps by the ssDNA-binding-deficient but ATRIP-binding proficient AA35-316 GST-APE1, or the ssDNA-binding-proficient ATRIP-binding-deficient AA101-316 GST-APE1 (Figure 3C). Furthermore, AA101-316 GST-APE1 was able to associate with ssDNA gaps but could not recruit endogenous ATRIP onto ssDNA in the presence of ssDNA-coated RPA (Figure 3D). All these data are in line with the model proposed in Figure 5B.

2. It is important to check the binding of ATRIP to ssDNA in the presence of both APE1 and RPA by checking the effect of differential ratios of APE1 and RPA on in vitro binding of ATRIP to the ssDNA as shown in Figures 1C and 3C. The authors can address whether the abilities of the two proteins to recruit ATRIP to the ssDNA is competitive or cooperative or independent.

We thank the suggestion. However, the suggested experiment would provide an insight into whether the RPA-dependent ATRIP recruitment and APE1-dependent ATRIP recruitment are competitive or cooperative, only when both RPA and APE1 don’t interact with each other. We have shown that APE1 interacts with RPA in vitro (Figure 4), and that RPA protein complex and APE1 protein can recruit ATRIP protein onto ssDNA gaps independently (Figure 1C and 3C). To test any potential relationship between the two types of ATRIP recruitment, we performed extra experiment to directly compare the ATRIP recruitment onto 80nt-ssDNA gaps in vitro when individual protein is present or when both proteins are present and added the description (Page 13, Line 364-377):

“We have shown that both RPA complex and APE1 protein can recruit recombinant ATRIP protein onto ssDNA gaps independently in vitro (Figures 1C and 3C), and that APE1 and RPA complex interact with each (Figure 4). It is interesting to test whether the RPA-dependent ATRIP recruitment to ssDNA and APE1-dependent ATRIP recruitment to ssDNA is competitive or cooperative. Using in vitro pulldown assays with beads coupled with 80nt-ssDNA gap, we found that the presence of both His-RPA protein complex and GST-APE1 protein increased the recruitment of His-ATRIP protein onto ssDNA gaps, compared with each protein individually (Lane 4 vs Lane 2/3, Figure 4—figure supplement 3E-3F). Interestingly, it seems there is no significance for the recruitment of His-ATRIP onto ssDNA when both proteins are present compared with the sum of when individual protein is present (Lane 4 vs (Lane 2 + Lane 3), Figure 4—figure supplement 3E-3F). Our observation suggests that the RPA-mediated and APE1-mediated ATRIP recruitment onto ssDNA is neither cooperative with each other nor competitive/exclusive to each other at least in in vitro reconstitution system under our experimental conditions.”

3. RBM1 in APE1 shows sequence similarity to an N-terminal region of ATRIP for RAP70N (OB-F of RPA70) binding. If so, the RBM1 of APE1 and the ATRIP region should show competitive bindings to RPA70N. This should be examined by competitive pull-down assay using purified APE1, ATRIP-N-terminal region, and RPA70N.

We thank the suggestion. We have shown in Figure 4 that RBM1 within APE1 is important for interaction with RPA complex and likely through RPA70. However, it has not been determined whether APE1 RBM1 is sufficient for the interaction with RPA70N. Second, AA1-100 His-APE1 containing the RBM1 motif is sufficient for the interaction with His-ATRIP in vitro (Figure 3A), suggesting that the NT100 domain of APE1 directly interacts with ATRIP. In addition, it remains controversial whether and how the N-terminal domain or other internal domains within ATRIP mediate its interaction with the RPA protein complex and/or RPA on ssDNA (Ball et al., 2005; Kim et al., 2005; Namiki and Zou, 2006). How APE1 NT100 motif engages ATRIP and RPA70 interactions and the recruitment of ATRIP onto ssDNA warrants further investigation. We re-wrote a paragraph relevant to the APE1 interaction with ATRIP and RPA in the Discussion section (Page 16, Line 454-469).

“It is an outstanding question in the field of ATR DDR regarding how exactly ATRIP protein is recruited onto ssDNA gaps in an RPA-dependent and/or -independent manner. We have elucidated in this study that the NT100 motif of APE1 is required and sufficient for ATRIP interaction in in vitro protein-protein interaction assays (Figure 3A), and that the APE1-ATRIP interaction is essential for the recruitment of recombinant ATRIP protein onto ssDNA gaps in in vitro reconstitution system and endogenous ATRIP protein to ssDNA gaps in *Xenopus* HSS system (Figure 3C-3D). Our observations support the critical role of the APE1-ATRIP interaction in the APE1-dependent ATRIP recruitment onto ssDNA gaps. For the Mode #2 APE1- and RPA-dependent recruitment of ATRIP onto ssDNA gaps, we hypothesize that an unknown Protein X may negatively regulate ATRIP binding to RPA complex especially RPA70 N-terminal domain, which has been involved in the recruitment of several DDR proteins to ssDNA such as ETAA1, Mre11, Nbs1, Rad9, p53, and PRIMPOL (Bhat and Cortez, 2018). APE1 protein may reverse or counteract with such inhibitory effect of Protein X in the recruitment of ATRIP onto ssDNA. Alternatively, APE1 NT100 motif interacts with ATRIP and RPA70, which may be required for the conformation change and/or stabilization of ATRIP-RPA-ssDNA in the *Xenopus* HSS system (Figure 5B). Future studies will test these different scenarios.”

4. Please demonstrate the in vivo role of APE1(APEX1) in ATR activation using siRNA inhibition in mammalian cells or APE1 deletion of yeast cells.

Done. Please see our response to the #2 Essential revisions.

5. Please show quantification of all blots with error bars with proper statistics (or claim on the reproducibility of the results)

Please see our response to the #3 Essential revisions.

Reviewer #2 (Recommendations for the authors):This reviewer suggests that the authors consider the following points to validate the model and increase the impact.1. The authors need to test the effect of RPA depletion (Figure 1, Figure 3, and Figure 4, at least one experiment) because DNA binding of ATRIP and ATR activation was always associated with RPA-DNA binding.

We have performed additional experiment of RPA depletion and shown the result in Figure 1—figure supplement 1A. Please see our detailed response to this seminar question in #2 Essential revisions.

2. It is not clear how PCNA is equally loaded on the gapped and non-gapped DNA substrates (Figure 1).

In Figure 1A, PCNA is used as a loading control for DNA-bound fractions for both control plasmid and gapped plasmid. The *Xenopus* HSS system was developed by Walter and Newport, in which both sperm chromatin DNA and dsDNA plasmid can form pre-replication complex (Walter and Newport, 2000; Walter et al., 1998). In particular, circular plasmid DNA undergoes Topoisomerase I-mediated topological changes and nucleosome formation in HSS system(Walter and Newport, 2000). This means wild type circular plasmid DNA can be cut into nicked plasmid DNA in HSS incubation. Later on, PCNA has been utilized by as loading controls for protein fraction isolation and comparison on plasmid DNA (wild type, nicked, gapped, or some damaged forms) by several *Xenopus* labs (Havens and Walter, 2009; Lin et al., 2018; Zembutsu and Waga, 2006).

3. The proposed model suggests that RPA prevents APE1 from ssDNA recognition (Figure 5). The significance of APE1 in ATR recruitment needs to be further discussed or experimentally addressed. RPA interacts weakly with short stretches of ssDNA. Does APE1 act specifically on a gapped DNA to stimulate the ATR pathway? Kim et al. (2005) showed that annealing of dA/dT oligos also triggers the ATR pathway. They could test whether APE1 also acts on annealed oligos. APE1 is an endonuclease that cleaves the DNA phosphodiester backbone at AP sites, leaving a 1 nt gap. Does APE1 activate the ATR pathway by DNA circles containing AP sites (Figure 1)?

We are not sure how the reviewer can tell our proposed model suggesting “that RPA prevents APE1 from ssDNA recognition (Figure 5).” To avoid potential confusion and misunderstanding, we revised our Figure 5 and illustrated possible mechanisms of ATRIP recruitment onto ssDNA gaps (30/80nt) that when added to the in vitro reconstitution system separately, RPA and APE1 protein can recruit recombinant ATRIP protein onto ssDNA gaps independently (Figure 5A), and that there are two modes of APE1-dependent RPA-dependent/independent recruitment of endogenous ATRIP protein onto ssDNA gaps in *Xenopus* egg extracts (Figure 5B). In addition, our data in Figure 4—figure supplement 3A-3D suggest that recombinant RPA protein complex increased the binding of APE1 protein onto ssDNA (40, 60, and 80nt), and that the RPA-APE1 interaction is important for the stimulated APE1 binding to ssDNA.

Our data in this study show that APE1 protein can associate with 30nt- and 80nt- ssDNA regions within defined gapped DNA structures and defined ssDNA structure (40, 60, and 80nt) (Figure 2). Consistent with our observations, other studies have shown that 22nt-ssDNA is sufficient for a stable hAPE1-ssDNA complex formation in in vitro EMSA assays and that hAPE1 associates with ssDNA regions at stalled DNA replication forks in vitro (Bazlekowa-Karaban et al., 2019; Hoitsma et al., 2022). Furthermore, mouse APE1 was shown to interact with 20nt-ssDNA and -dsDNA for exonucleolytic cleavage in vitro (Liu et al., 2021). Although we can’t exclude the possibility that APE1 may associate with other distinct DNA and/or RNA structures, our findings in this manuscript suggest that APE1 recognizes ssDNA and recruits ATRIP (and potentially the tight ATR/ATRIP complex) onto ssDNA for the activation of ATR DDR pathway.

The ATR DDR pathway activation by the annealing synthetic oligonucleotides dA_70_/dT_70_ in *Xenopus* egg extracts was first characterized by the Dunphy group in 2000 (Kumagai and Dunphy, 2000). However, later studies have shown that the model dA_70_/dT_70_ system also triggers ATM DDR activation, which requires TopBP1 and the MRN complex (Yoo et al., 2007; Yoo et al., 2009). Furthermore, the exact structures of the model dA_70_/dT_70_ system (e.g., 70bp dsDNA with blunt end, 5’-junction ssDNA/dsDNA, 3’-junction ssDNA/dsDNA, or mixtures of these different forms) remains to be determined after 23-year studies. To visualize the mobility shift of phosphorylated version of Chk1, phosphatase inhibitor tautomycin is typically added to the *Xenopus* egg extracts (Kim et al., 2005; Kumagai and Dunphy, 2000), which makes the observations using the model dA_70_/dT_70_ system more difficult to be interpreted regarding the direct and/indirect effect. Our study aims to use the more defined 30nt- and 80nt-ssDNA gap structures and determine the role and mechanism of APE1 in the ATR DDR pathway activation. Whether and how APE1 contributes to the ATR and/or ATM DDR pathway using model dA_70_/dT_70_ system in *Xenopus* system is determined to be out of the scope of current study; however, it may be interesting to test this in the future studies.

Our current study focuses on how APE1 interacts and ssDNA and recruits ATRIP onto ssDNA via a non-catalytic function. “Does APE1 activate the ATR pathway by DNA circles containing AP sites?” In our opinion, it is very likely. One of our recently published studies using *Xenopus* HSS system has determined that APE1 and its 3’-5’ exonuclease activity is important for the ATR-Chk1 DDR pathway activation in response to the defined plasmid-based SSB structures (Lin et al., 2020). Future studies using plasmid DNA containing site-specific AP site may be utilized to test whether APE1 and its AP endonuclease activity is critical for the defined AP site-derived ATR/Chk1 DDR pathway activation. In the first and second paragraph of the Discussion section (Page 14), we did add more description regarding the current understanding of APE1 and its nuclease activity in the DDR pathway activation: “In addition to its critical roles in DNA repair and redox regulation (Li and Wilson, 2014; Tell et al., 2009), accumulating evidence suggests important roles of APE1 in the activation of the ATR-Chk1 DDR pathway (Li et al., 2022; Lin et al., 2020; Vrouwe et al., 2011). We and others have demonstrated that APE1 and its nuclease activity are important for the ATR-Chk1 DDR activation in response to oxidative DNA damage and ultraviolet-damage in mammalian cells (Li et al., 2022; Vrouwe et al., 2011). Furthermore, APE1 plays an essential role in the initiation step of 3'-5' end resection in SSB-induced ATR-Chk1 DDR pathway via its 3'-5' exonuclease acitivity in *Xenopus* egg extract system (Lin et al., 2020). Mechanistic studies further elucidate that APE1 directly recognizes and binds to SSB site and generate a small ssDNA gap structure via its catalytic function for the subsequent APE2 recruitment and activation for the continuation of SSB end resection (Lin et al., 2020). Notably, APE1 forms biomolecular condensates in vitro and in nucleoli independent of its nuclease activity to promote the ATR-Chk1 DDR pathway activation in cancer cells, and APE1 is proposed as a new direct activator of the ATR kinase, in addition to TopBP1 and ETAA1 (Li and Yan, 2023; Li et al., 2022)…..In current study, we have identified and characterzied another distinct regulatory mechanism of APE1 in the ATR-Chk1 DDR pathway independent of its nuclease activity.”

Reviewer #3 (Recommendations for the authors):1. The title is misleading. As depicted in the model in Figure 5B, the main role of APE1 in promoting ATR activation in HSS extracts seems to be to counteract a putative inhibitor of ATRIP recruitment. However, there is no data in the paper showing that this role is linked to the RPA-independent mechanism of ATRIP recruitment. It is suggested to remove "RPA-independent manner" from the title.

We have shown in our initial submission that with the absence of each other, recombinant RPA protein complex and APE1 protein can recruit ATRIP protein onto ssDNA gaps independently in in vitro reconstitution system (Figure 1C and 3C). We have also shown that APE1 depletion led to compromised ATRIP recruitment onto ssDNA gaps (Figure 1B) and that RPA-depletion led to decreased ATRIP recruitment onto ssDNA gaps in *Xenopus* HSS system (Figure 1—figure supplement 1A). All these observations suggest that APE1 mediates ATRIP onto ssDNA in an RPA-dependent and RPA-independent manner. Therefore, we revised our model in new Figure 5 showing the RPA-dependent and APE1-dependent recruitment of ATRIP onto ssDNA gaps in in vitro reconstitution system (Figure 5A) and the APE1-dependent recruitment of endogenous ATRIP protein onto ssDNA gaps in RPA-dependent and -independent manner in *Xenopus* egg extracts (Figure 5B). To better reflect the major findings in this study, we also revised the title of our manuscript to “APE1 recruits ATRIP to ssDNA in an RPA-dependent and -independent manner to promote the ATR DNA damage response”.

2. An important experiment missing is to test the effect of RPA on APE1-dependent ATRIP recruitment. In the in vitro experimental setting, the authors should compare absence versus presence of RPA to address if the addition of RPA strongly enhances ATRIP recruitment when APE1 is present.

We have shown that in the absence of each other, recombinant RPA protein complex and APE1 protein can recruit ATRIP protein onto ssDNA gaps independently in in vitro reconstitution system (Figure 1C and 3C). We did perform additional in vitro experiment in which purified recombinant GST-APE1 protein, His-RPA complex, or both proteins were present and found that APE1-mediated ATRIP recruitment and RPA-mediated ATRIP recruitment are neither competitive nor cooperative, but rather are independent from each other (Figure 4 —figure supplement 3E-3F).

3. The authors never discuss the importance of APE1 for ATR activation in different contexts and different lesion types. Is the role of APE1 in promoting ATRIP recruitment broadly required for ATR activation? If so, one would predict that cells lacking APE1 should exhibit deficient ATR signaling in response to several types of genotoxins (and is that the case?). If not, then how do the authors explain the central importance of APE1 in ATRIP recruitment in their system given that ssDNA is a general trigger of ATR signaling?

We have included additional experiment in the revised manuscript showing that siRNA-mediated knockdown of human APE1 compromised the hydrogen peroxide-induced ATR/Chk1 DDR pathway activation in human cultured cancer cell U2OS cells (Figure 1—figure supplement 1B). Consistent with the significant mechanism of APE1 in ATRIP recruitment onto ssDNA in ATR DDR, previous studies have shown the APE1 is important for the ATR DDR activation in response to oxidative stress, alkylation damage, and UV-damage in culture mammalian cells (Li et al., 2022; Vrouwe et al., 2011). More detailed description can be found in our response to #2 Essential Revisions.

4. In Figures 1A and 1B, APE1 depletion completely blocked ATRIP recruitment to Gap plasmids and dsDNA containing 30nt and 80nt ssDNA gaps. These results indicated that APE1 is essential for the recruitment of ATRIP onto RPA-coated ssDNA in *Xenopus* HSS and RPA cannot bind ATRIP without the presence of APE1. However, this explanation conflicts with the proposed model in Figure 5A, in which RPA is able to recruit ATRIP without APE1. As such, the working model in Figure 5A is not supported by the HSS experiments performed in Figure 1. The authors need to clarify this point of confusion, which is probably attributed to the model in Figure 5A being based only on in vitro data.

Thanks for the question. To make better clarification and avoid potential misunderstanding, we revised our Figure 5 and illustrated the RPA-dependent and APE1-dependent recruitment of ATRIP onto ssDNA gaps “In in vitro reconstitution system” (Figure 5A), and the two modes of APE1-dependent recruitment of ATRIP protein onto ssDNA gaps “in *Xenopus* egg extracts” (Figure 5B).

5. In Figure 1D the authors claim that APE1 increases the recruitment of ATRIP, however, such an increase in recruitment is not obvious from the presented blot, and careful quantitation (with replicates and statistics) should be presented to support the claim.

The intensity of ATRIP onto ssDNA gap structures were quantified and statistical analysis shows the presence of GST-APE1 protein increased the recruitment of His-ATRIP protein onto ssDNA gap structures, compared with GST (Lane 5 vs Lane 2, Lane 6 vs Lane 3, Figure 1D).

6. In Figures 2B and 2C, the authors show that APE1-ssDNA association occurs in a length-dependent manner in vitro, where no ssDNA shorter than 30nt is shown to interact with APE1. However, this is in conflict with Figure 1A, in which Gap plasmid resected to reveal 18-26nt of ssDNA (refer to line 129) is still able to robustly interact with APE1. The authors should address this discrepancy.

Our data in this study show that APE1 protein can associate with 30nt- and 80nt- ssDNA regions within defined gapped DNA structures and defined ssDNA structure (40, 60, and 80nt) (Figure 2). Consistent with our observations, other studies have shown that 22nt-ssDNA and 28nt-ssDNA are sufficient for a stable hAPE1-ssDNA complex formation in in vitro EMSA assays and that hAPE1 associates with ssDNA regions at stalled DNA replication forks in vitro (Bazlekowa-Karaban et al., 2019; Hoitsma et al., 2022). These observations are in line with our estimated ssDNA gap size (~18-26nt) through the distinct 3’-5’ SSB end rection for ATR DDR activation in response to defined SSB structures in the *Xenopus* HSS system (Lin et al., 2018; Lin et al., 2020). It is worth noting that the estimated ssDNA gaps (~18-26nt) at defined SSB site did not include the possible 5’-3’ SSB end rection in *Xenopus* HSS system. More description of ssDNA gap size has been included in the second paragraph in the Discussion section (Page 15, Line 433-446).

7. The concentrations of APE1 proteins used for EMSA assay should be disclosed to show the binding affinity of APE1 and ssDNA. Meanwhile, whether this affinity is physiologically relevant?

Thanks for the suggestion, and we have added the concentrations of APE1 protein (1, 5, and 20 μM) used for the EMSA assays in Figure 2E.

“Whereas the concentration of *Xenopus* APE1 protein in *Xenopus laevis* egg was estimated to be ~1.5 μM (Wuhr et al., 2014), the concentration of hAPE1 protein in HEK293T cells was estimated to ~2.8 μM (https://opencell.czbiohub.org/gene/ENSG00000100823) (Wisniewski et al., 2014). Considering most APE1 protein is localized inside of the nucleus of mammalian cells (Li and Wilson, 2014), the observed APE1 affinity and interaction with ssDNA in our in vitro EMSA assays (Figure 2E) are physiologically relevant.”.

This part has been added to the Discussion section (Page 15, Line 426-431).

8. In Figure 4A, the pulldown of GST-APE1 AA1-34 still pulls down RPA70 even though both RPA-binding motifs are entirely absent. This conflicts with the result seen in the double RBM mutant, where no RPA is pulled down at all in Figure 4C. The authors should address this discrepancy.

We run another two independent experiments (Repeat 1 and Repeat 2). From the three independent experiments (see the blots of RPA70 bands Author response image 1), we chose Repeat 1 as representative results in our revised figures (New Figure 4A). Our quantification of RPA70 bands from pulldown samples normalized to RPA70 bands from input indicated that AA1-34 APE1 displayed significantly decreased capacity to interact with His-RPA protein, compared with WT APE1 (New Figure 4—figure supplement 1A). We recognize that there are possible differences between point mutations and protein fragments. We can’t completely exclude the possibility that the AA1-34 fragment of APE1 may have another alternative binding site for RPA complex.

**Author response image 1. sa2fig1:** IB against anti-RPA in pulldown samples.

9. Overall, writing should be improved throughout the manuscript.

Yes, we revised the manuscript accordingly.

**Reference used in this response:**

Ball, H.L., J.S. Myers, and D. Cortez. 2005. ATRIP binding to replication protein A-single-stranded DNA promotes ATR-ATRIP localization but is dispensable for Chk1 phosphorylation. *Mol Biol Cell*. 16:2372-2381.

Bazlekowa-Karaban, M., P. Prorok, S. Baconnais, S. Taipakova, Z. Akishev, D. Zembrzuska, A.V. Popov, A.V. Endutkin, R. Groisman, A.A. Ishchenko, B.T. Matkarimov, A. Bissenbaev, E. Le Cam, D.O. Zharkov, B. Tudek, and M. Saparbaev. 2019. Mechanism of stimulation of DNA binding of the transcription factors by human apurinic/apyrimidinic endonuclease 1, APE1. *DNA Repair (Amst)*. 82:102698.

Bhat, K.P., and D. Cortez. 2018. RPA and RAD51: fork reversal, fork protection, and genome stability. *Nat Struct Mol Biol*. 25:446-453.

Cimprich, K.A., and D. Cortez. 2008. ATR: an essential regulator of genome integrity. *Nat Rev Mol Cell Biol*. 9:616-627.

Havens, C.G., and J.C. Walter. 2009. Docking of a specialized PIP Box onto chromatin-bound PCNA creates a degron for the ubiquitin ligase CRL4Cdt2. *Mol Cell*. 35:93-104.

Hoitsma, N.M., J. Norris, T.H. Khoang, V. Kaushik, E. Antony, M. Hedglin, and B.D. Freudenthal. 2022. Mechanistic Insight into AP-Endonuclease 1 Cleavage of Abasic Sites at Stalled Replication Forks. *bioRxiv preprint*.

Kim, S.M., A. Kumagai, J. Lee, and W.G. Dunphy. 2005. Phosphorylation of Chk1 by ATM- and Rad3-related (ATR) in *Xenopus* egg extracts requires binding of ATRIP to ATR but not the stable DNA-binding or coiled-coil domains of ATRIP. *J Biol Chem*. 280:38355-38364.

Kumagai, A., and W.G. Dunphy. 2000. Claspin, a novel protein required for the activation of Chk1 during a DNA replication checkpoint response in *Xenopus* egg extracts. *Mol Cell*. 6:839-849.

Li, J., and S. Yan. 2023. Molecular mechanisms of nucleolar DNA damage checkpoint response. *Trends Cell Biol*. 33:361-364.

Li, J., H. Zhao, A. McMahon, and S. Yan. 2022. APE1 assembles biomolecular condensates to promote the ATR-Chk1 DNA damage response in nucleolus. *Nucleic Acids Res*. 50:10503-10525.

Li, M., and D.M. Wilson, 3rd. 2014. Human apurinic/apyrimidinic endonuclease 1. *Antioxid Redox Signal*. 20:678-707.

Lin, Y., L. Bai, S. Cupello, M.A. Hossain, B. Deem, M. McLeod, J. Raj, and S. Yan. 2018. APE2 promotes DNA damage response pathway from a single-strand break. *Nucleic Acids Res*. 46:2479-2494.

Lin, Y., J. Raj, J. Li, A. Ha, M.A. Hossain, C. Richardson, P. Mukherjee, and S. Yan. 2020. APE1 senses DNA single-strand breaks for repair and signaling. *Nucleic Acids Res*. 48:1925-1940.

Liu, T.C., C.T. Lin, K.C. Chang, K.W. Guo, S. Wang, J.W. Chu, and Y.Y. Hsiao. 2021. APE1 distinguishes DNA substrates in exonucleolytic cleavage by induced space-filling. *Nat Commun*. 12:601.

Marechal, A., and L. Zou. 2015. RPA-coated single-stranded DNA as a platform for post-translational modifications in the DNA damage response. *Cell Res*. 25:9-23.

Namiki, Y., and L. Zou. 2006. ATRIP associates with replication protein A-coated ssDNA through multiple interactions. *Proc Natl Acad Sci U S A*. 103:580-585.

Tell, G., F. Quadrifoglio, C. Tiribelli, and M.R. Kelley. 2009. The many functions of APE1/Ref-1: not only a DNA repair enzyme. *Antioxid Redox Signal*. 11:601-620.

Vrouwe, M.G., A. Pines, R.M. Overmeer, K. Hanada, and L.H. Mullenders. 2011. UV-induced photolesions elicit ATR-kinase-dependent signaling in non-cycling cells through nucleotide excision repair-dependent and -independent pathways. *J Cell Sci*. 124:435-446.

Walter, J., and J. Newport. 2000. Initiation of eukaryotic DNA replication: origin unwinding and sequential chromatin association of Cdc45, RPA, and DNA polymerase α. *Mol Cell*. 5:617-627.

Walter, J., L. Sun, and J. Newport. 1998. Regulated chromosomal DNA replication in the absence of a nucleus. *Mol Cell*. 1:519-529.

Wisniewski, J.R., M.Y. Hein, J. Cox, and M. Mann. 2014. A "proteomic ruler" for protein copy number and concentration estimation without spike-in standards. *Molecular & cellular proteomics : MCP*. 13:3497-3506.

Wuhr, M., R.M. Freeman, Jr., M. Presler, M.E. Horb, L. Peshkin, S.P. Gygi, and M.W. Kirschner. 2014. Deep proteomics of the *Xenopus laevis* egg using an mRNA-derived reference database. *Curr Biol*. 24:1467-1475.

Yoo, H.Y., A. Kumagai, A. Shevchenko, and W.G. Dunphy. 2007. Ataxia-telangiectasia mutated (ATM)-dependent activation of ATR occurs through phosphorylation of TopBP1 by ATM. *J Biol Chem*. 282:17501-17506.

Yoo, H.Y., A. Kumagai, A. Shevchenko, and W.G. Dunphy. 2009. The Mre11-Rad50-Nbs1 complex mediates activation of TopBP1 by ATM. *Mol Biol Cell*. 20:2351-2360.

Zembutsu, A., and S. Waga. 2006. de novo assembly of genuine replication forks on an immobilized circular plasmid in *Xenopus* egg extracts. *Nucleic Acids Res*. 34:e91.

Zou, L., and S.J. Elledge. 2003. Sensing DNA damage through ATRIP recognition of RPA-ssDNA complexes. *Science*. 300:1542-1548.